# Induction of Rosette-to-Lumen stage embryoids using reprogramming paradigms in ESCs

Jan Langkabel[1,9], Arik Horne[2,3,9], Lorenzo Bonaguro [2,3], Lisa Holsten [2,3], Tatiana Hesse[1], Alexej Knaus[1,4], Yannick Riedel[1], Matthias Becker [3], Kristian Händler[2,5], Tarek Elmzzahi[6,7], Kevin Bassler[2], Nico Reusch [2], Leon Harootoonovtch Yeghiazarian[2], Tal Pecht[2], Adem Saglam[3,5], Thomas Ulas[2,3,5], Anna C. Aschenbrenner[2,3,5,8], Franziska Kaiser[1], Caroline Kubaczka [1], Joachim L. Schultze[2,3,5,10] & Hubert Schorle [1,10 ✉]

Blastocyst-derived stem cell lines were shown to self-organize into embryo-like structures in 3D cell culture environments. Here, we provide evidence that embryo-like structures can be generated solely based on transcription factor-mediated reprogramming of embryonic stem cells in a simple 3D co-culture system. Embryonic stem cells in these cultures self-organize into elongated, compartmentalized embryo-like structures reflecting aspects of the inner regions of the early post-implantation embryo. Single-cell RNA-sequencing reveals transcriptional profiles resembling epiblast, primitive-/visceral endoderm, and extraembryonic ectoderm of early murine embryos around E4.5–E5.5. In this stem cell-based embryo model, progression from rosette formation to lumenogenesis accompanied by progression from naïve- to primed pluripotency was observed within Epi-like cells. Additionally, lineage specification of primordial germ cells and distal/anterior visceral endoderm-like cells was observed in epiblast- or visceral endoderm-like compartments, respectively. The system presented in this study allows for fast and reproducible generation of embryo-like structures, providing an additional tool to study aspects of early embryogenesis.

[1] Institute of Pathology, Department of Developmental Pathology, University Hospital Bonn, University of Bonn, Bonn, Germany. [2] Genomics and Immunoregulation, Life and Medical Sciences (LIMES) Institute, University of Bonn, Bonn, Germany. [3] Systems Medicine, DZNE, Bonn, Germany and University of Bonn, Bonn, Germany. [4] Institute for Genomic Statistics and Bioinformatics, University Hospital Bonn, University of Bonn, Bonn, Germany. [5] PRECISE Platform for Single Cell Genomics and Epigenomics, Deutsches Zentrum für Neurodegenerative Erkrankungen (DZNE), Bonn, Germany and University of Bonn, Bonn, Germany. [6] Molecular Immunology in Neurodegeneration, DZNE, Bonn, Germany. [7] Department of Microbiology and Immunology, The Peter Doherty Institute for Infection and Immunity, University of Melbourne, Melbourne, VIC, Australia. [8] Department of Internal Medicine and Radboud Center for Infectious Diseases (RCI), Radboud University Medical Center, Nijmegen, The Netherlands. [9] These authors contributed equally: Jan Langkabel, Arik Horne. [10] These authors jointly supervised this work: Joachim L. Schultze, Hubert Schorle. ✉email: schorle@uni-bonn.de

Cell-culture-based embryo models that can recapitulate embryogenesis or specific developmental hallmarks are currently gaining increasing interest in the scientific community, as they provide tools to study mammalian embryogenesis in vitro. Mammalian embryonic development begins with the fertilized egg, which develops into the blastocyst at embryonic day (E) 3.5, comprising of three lineages: The trophectoderm (TE) surrounding the inner-cell-mass (ICM), which is lined by the primitive endoderm (PE), separating the ICM from the blastocoel[1]. From blastocysts, three stem cell lineages can be derived and propagated in cell culture indefinitely: trophoblast stem cells (TSCs)[2], embryonic stem cells (ESCs)[3], and extra-embryonic endoderm (XEN) stem cells[4]. The implantation of the blastocyst into the uterus is mediated by the mural TE, that gives rise to primary trophoblast giant cells (TGCs) and subsequently becomes covered by the Reichert's membrane and the parietal endoderm, derived from the primitive endoderm[5]. After implantation, the developing embryo forms a cylindrical structure, in which the ICM-derived epiblast (Epi) lies distally within the conceptus, lined proximally by the TE-derived extra-embryonic ectoderm (ExE)[5]. Both tissues are surrounded by the PE derived visceral endoderm (VE), the VE lining the Epi is referred to as embryonic VE (EmVE), while the VE adjacent to the ExE is called extraembryonic VE (ExVE)[6]. Emerging from the ExE, the ectoplacental cone builds the most proximal part of the conceptus, thereby establishing the maternal-fetal interface after implantation[7].

It was previously demonstrated that the three types of blastocyst-derived stem cells (ESCs, TSCs, and XEN cells) can organize into early embryo-like structures, forming tissues resembling early inner embryonic architecture, when co-cultured in 3D cell culture environments[8,9]. These structures exhibit key hallmarks of embryogenesis, such as patterning events and they were demonstrated to implant in uteri upon transplantation[9]. However, as implantation occurs at the blastocyst stage, mediated by the trophectoderm, which is not formed in the previously described embryo model[9], the question has to be raised whether the observed implantations are in fact uterine reactions, induced by mechanical stimuli of the uterus. Nevertheless, all of these embryo model systems can be used to recapitulate specific developmental events that occur during early embryogenesis and can potentially increase our knowledge in processes and diseases involved in embryogenesis. As a rapidly evolving field of research the potentials and limitations of such embryo-like organoids are currently actively discussed in the scientific community. The generation of these embryo-like structures relies on complex cell culture requirements for the maintenance of each of the three stem cell types. Due to differences in proliferation and cell cycle, precise timing is imperative for the success of generating such stem-cell-based embryo models. Recently, Amadei et al.[10] presented induced ETX (iETX) embryos, which can be generated from a starting population of TSCs and two ESC lines, one of which can be reprogrammed towards a VE-like identity by overexpression of Gata4[10]. This two stem-cell-based system displays significantly improved developmental potential compared to previously presented ETX embryos[8].

Here, we report an alternative approach for the generation of stem-cell-based embryo models, using a transcription factor-mediated cellular reprogramming regimen of genetically manipulated ESCs to give rise to induced trophoblast stem cell- (iTSC) or induced extraembryonic endoderm stem cell- (iXEN) identities. Thus, we eliminate the need for multiple, individual stem cell cultures (ESC, TSC, and XEN cells) as well as complex cell culture reagents. Using an easy to generate agarose-based 3D-culture system, we demonstrate that transgene-induced cellular reprogramming and self-organization into embryo-like tissues

occur in parallel. The resulting structures resemble the inner regions of early post-implantation murine embryos at E4.5–E5.5. Single-cell RNA sequencing (scRNA-seq) reveals that the transcriptional profiles of the three tissues are highly similar to their respective embryonic counterpart, exhibiting molecular hallmarks of natural development. Furthermore, morphological, and molecular hallmarks, such as the formation of distal visceral endoderm (DVE)/anterior visceral endoderm (AVE), initiation of pluripotency state progression from naïve- to primed-pluripotency, as well as primordial germ cell specification (PGC) with the required tissue interaction was observed.

## Results

**Reprogramming in co-culture induces embryo-like architecture.** In recent years, we and others demonstrated that ESCs can be reprogrammed to bona fide TSCs and XEN cells[11–14]. Here, reprogramming of ESC to iTSC was achieved by induction of transgenes encoding for Cdx2, Tfap2c, Eomes, Gata3, and Ets2[14], and reprogramming of ESC to iXEN cells is possible by overexpression of Gata6[12,13,15]. To test whether such transcription factor-mediated cellular reprogramming in defined ESC lines is sufficient to generate embryo-like structures, we used our 5 Factor ESC line (5F-ESC) carrying doxycycline-inducible transgenes of Cdx2, Tfap2c, Eomes, Gata3, and Ets2[14] in combination with an ESC line carrying a doxycycline-inducible Gata6 transgene (iGATA6 ESC)[15] in addition to an ESC line carrying an Pou5f1 (alias: Oct3/4) promoter-driven GFP cassette, termed here 'Kermit ESC'. While 5F-ESCs give rise to iTSC[14], iGATA6 ESCs were predicted to reprogram to iXEN[12,15] and Kermit ESCs to ICM derivatives. All cells were grown under standard ESC culture conditions (FCS, 2i/LIF). To initiate the generation of embryo-like structures, the three ESC lines were co-cultured in agarose micro-tissue wells to enable non-adherent growth in 3D (Fig. 1a). After 24 h culture in standard ESC medium, the culture medium was switched to reconstructed embryo medium[9] supplemented with 2 µg/ml doxycycline (DOX), to induce transgene expression of 5F-ESCs and iGATA6 ESCs, initiating reprogramming into an iTSC- or iXEN cell fate, respectively. We observed that the cell aggregates underwent morphological changes and self-organized into elongated and compartmentalized structures (Fig. 1a, b). In stark contrast, without the addition of DOX, aggregates of the three ESC lines display a 'salt-and-pepper-like' distribution throughout the culture period (Supplementary Fig. 1a). This result indicated that transgene induction over 3 days in combination with aggregation culture leads to a self-propelled separation and reorganization of the cells in the aggregates. After 3 days, the expression of transgenes was stopped by omitting DOX. Then, 24 h after depletion of DOX, the aggregates showed compartmentation into an epiblast (Epi)-like compartment consisting of (GFP+) Kermit ESCs-derived cells, adjacent to a CDX2-positive extraembryonic ectoderm (ExE)-like compartment, most likely derived from 5F-ESCs, and a GATA4-positive sphere-like structure surrounding the two inner compartments, resembling a visceral endoderm (VE)-like compartment (Fig. 1b and Supplementary Fig. 1b, c). Since it is known that GATA6 induces expression of Gata4[13], we propose the VE-like compartment to be descendants of the iGATA6 ESCs. We next tested different cell ratios of the three ESC lines and found that an averaged combination of 6 Kermit ESCs, 16 5F-ESCs, and 5 iGATA6 ESCs per microwell of the agarose 3D-culture dish yielded the highest number of correctly compartmented structures. This is in line with cell ratios recently used for the generation of ETX embryos[8]. In three independent experiments, we counted a total of 1167 aggregates (day 4) and 778 aggregates (day 5) of which ~25% displayed correct compartmentation 5 days after seeding,

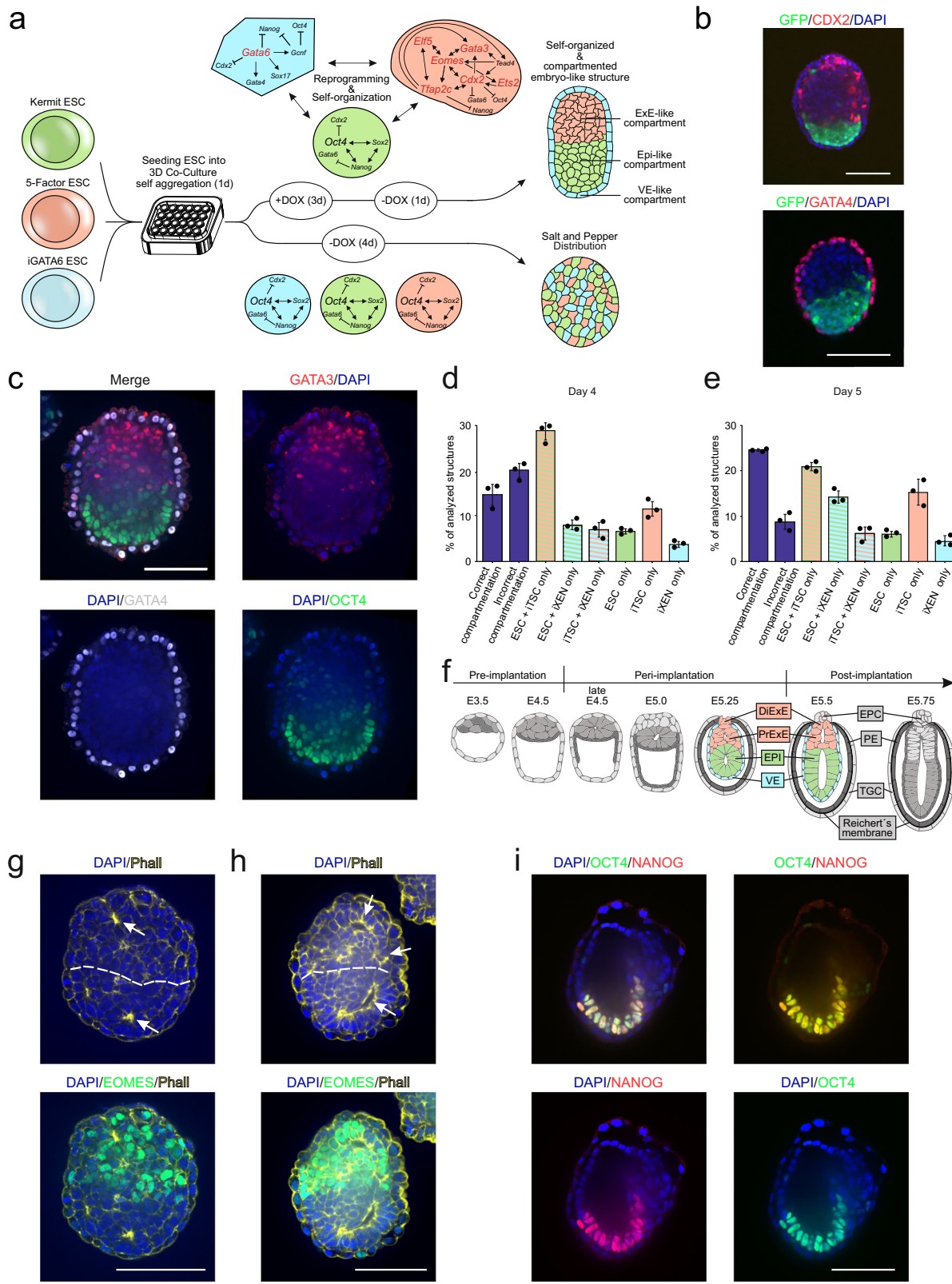

suggesting successful reprogramming and self-organization (Fig. 1d, e). Of note, this is comparable to efficiency rates reported for other embryo model approaches, like ETX embryos (29.8%)[8], iETX embryos (between 20 and 30%)[10], and ETX-embryoids (~23%)[9].

Having established a general protocol for the induction of such embryo-like structures by reprogramming paradigms, we substituted Kermit ESCs with an unmodified, non-fluorescent ESC line, KNUT1[16], which allowed for a more detailed analysis of marker expression and patterning within the Epi-like compartment. These embryo-like structures were subjected to immunofluorescence (IF) staining against OCT4, GATA3, and GATA4, and again revealed high organizational capacity of the structures generated, showing an OCT4 + Epi-like compartment next to a

**Fig. 1 Generation of embryo-like structures using reprogramming paradigms in ESCs. a** Schematic protocol for the generation of embryo-like structures following the method introduced in this study. The three starting cell lines are seeded in 3D Petri Dishes and are allowed to settle and aggregate for 24 h. Thereafter transgene expression in 5 Factor and iGATA6 ESCs is initiated by culture medium supplementation with DOX for three days. After an additional day of culture without DOX, aggregates have formed VE-, Epi-, and ExE-like cellular compartment, resembling early, inner embryonic architecture. **b** Self-assembled embryo-like structures showing early embryo architecture comprised of a OCT4+(GFP+; green) Epi-like compartment, next to a CDX2+ (red) ExE-like compartment, both of which are surrounded by a GATA4 + (red) VE-like compartment (lower image). DAPI, blue. **c** Substitution of Kermit ESCs with KNUT1 ESC, a non-fluorescent, unmodified ESC line resulted in similar self-organization, displaying restriction of Epi- (OCT4+; green), ExE- (GATA3+; red), and VE- (GATA4+; gray) marker expression to their respective compartments. DAPI, blue. **d** At Day 4 into the protocol (Day of DOX Stop) ~15% (total $n = 1167$) of the structures analyzed displayed embryo-like architecture. **e** At Day 5 into the protocol ~25% (total $n = 778$) of the structures analyzed displayed embryo-like architecture. Data of **d** and **e** are presented as mean values ± SD. Source data of **d** and **e** are provided as a Source Data file. **f** Scheme of embryonic architecture of E3.5 to E5.75 embryos. EPI-, ExE-, and VE-like compartments resemble the inner compartments of murine embryos between E5.25 and E5.5. DiExE Distal ExE, PrExE Proximal ExE, EPC ectoplacental cone, PE parietal endoderm, TGC trophoblast giant cell. **g** RtL-embryoids displayed rosette formation, indicated by aggregation of actin filaments. **h** RtL-embryoids displayed lumen formation in Epi- and ExE-like compartments. Phalloidin (Phall), yellow; EOMES, green; DAPI, blue. **i** In rare cases, complete lumenogenesis was observed, showing formation of a cavity spanning both Epi- and ExE-like compartments. Scale bars = 100 μm; Dotted lines indicate border of Epi- and ExE-like compartments. OCT4, green; NANOG, red; DAPI, blue. White arrows indicate rosette and lumen formation. Experiments were repeated independently at least three times with similar results (**b**, **c**, **g**, **h**, **i**).

---

GATA3+ ExE-like compartment, both of which were detected to be surrounded by a GATA4+ VE-like layer (Fig. 1c). As we observed the formation of organized tissues, resembling the embryonic architecture of the inner, embryonic, and extraembryonic tissues around E5.5, we next assessed the potential of rosette and lumen formation. During embryogenesis, rosettes are formed in both, Epi and ExE, which progress to form lumina within the two inner compartments. Ultimately, the two lumina fuse, generating the pro-amniotic cavity (Fig. 1f). Our Embryo-like structures displayed rosette formation in both Epi- and ExE-like compartments, as indicated by aggregation of actin fibers, which further progressed forming lumen within the respective compartments (Fig. 1g, h). We therefore termed the embryo-like structures Rosette-to-Lumen-embryoids (RtL-embryoids), as this describes the embryonic developmental stage the structures mimic in vitro. In rare cases we observed fusion of the lumen, forming structures that seemingly progressed to a more advanced developmental stage, resembling ~E6.5 embryos (Fig. 1i). While we were unable to further characterize such E6.5-like structures due to their rare appearance, this observation provides evidence of further developmental potential of such RtL-embryoids. Aggregates composed of ESCs and cells reprogrammed from iGATA6 ESCs also readily displayed lumen formation in the center of the ESC compartment, similar to ExE-embryoids published by Zhang et al.[9], that consist of blastocyst-derived ESCs and XEN cells (Supplementary Fig. 1d).

**Distinct transcriptional profiles of embryo-like tissues.** After observing self-organization into an embryo-like architecture, we used scRNA-Seq to analyze the transcriptomes of the cells within RtL-embryoids. We were interested to determine whether the three compartments immunohistochemically defined are characterized by three distinct cell lineages and whether there is further cellular heterogeneity within each compartment. To enable a preselection of correctly compartmentalized structures, RtL-embryoids were assembled using a mCherry-transduced iGATA6 ESC line. This allowed for the manual preselection of aggregates showing a mCherry+ sphere-like structure surrounding a GFP + compartment consisting of Kermit ESCs, adjacent to an unstained ExE-like compartment (Supplementary Fig. 1c). Incorrectly assembled or incomplete structures (Supplementary Fig. 1e, f) could thereby be excluded from the subsequent scRNA-Seq analysis. Twenty four hours after depletion of DOX, RtL-embryoids were harvested and homogenized to a single-cell suspension. Staining against CD40 in combination with GFP-signal-based exclusion of Epi-like cells allowed to

separate CD40 + ExE-like cells from the remainder of the embryo-like structures, as described for early murine embryos[17]. Together with GFP and mCherry, this labeling allowed for the identification of VE-like cells (mCherry + /GFP−/CD40−), Epi-like cells (GFP + /mCherry−/CD40−), and ExE-like cells (CD40 + /GFP−/mCherry−). To obtain high-quality single-cell transcriptomes, each cell type was sorted proportionally into 384-well plates before SMART-seq2 library preparation[18] (Fig. 2a and Supplementary Fig. 2a). A total of 961 cells passed our quality criteria (see method section). Uniform Manifold Approximation and Projection (UMAP) revealed a clear separation of three distinct cell clusters (Fig. 2b). To classify the cell types within each cluster we assessed expression of known marker genes for Epi, ExE, and VE and identified cluster 1 to be VE-like cells (*Amn*, *Dkk1*, *Gata4*, and *Sox17*), cluster 2 as Epi-like cells (*Pou5f1*, *Nanog*, *Gdf3*, and *Tdgf1*) and cluster 3 as ExE-like cells (*Cdx2*, *Elf5*, *Eomes*, and *Tfap2c*) (Fig. 2c and Supplementary Fig. 2f). These results showed that lineage-specific reprogramming had occurred in the three compartments previously identified by immunohistochemistry. Analyzing typical quality parameters in scRNA-Seq data revealed a comparable number of genes per cell, uniquely aligned genetic reads, cell numbers, and read variation between the three clusters (Supplementary Fig. 2b–e). Next, we used the FindAllMarkers() Seurat differential expression analysis using the default two-sided nonparametric Wilcoxon rank sum test with Bonferroni correction using all genes in the dataset and selected the top 100 differentially expressed genes (DEG) between the three clusters and performed GO term enrichment analysis, which revealed enrichment in 'extracellular structure formation' and 'endoderm development' terms in cells of the VE-like cluster, further supporting their successful transcriptional reprogramming (Fig. 2d). Cells within the Epi-like cluster were characterized by terms including 'response to leukemia inhibitory factor' and 'gastrulation'. Among the highest enriched gene sets in the ExE-like cluster were the 'regulation of actin-filament-based process' and 'epithelial cell development', which hints at a more differentiated ExE-like compartment. However, we detected enrichment sets like 'placenta-' and 'embryonic placenta development', suggesting specific developmental programs in this cellular compartment. To determine upstream regulators of the transcriptionally reprogrammed ESCs we performed transcription factor binding prediction (TFBP) analysis (Supplementary Fig. 2g). We observed a strong enrichment of the transcription factor *Sall4* within cells of the VE-like cluster, which is a gene encoding for the key regulator of the XEN lineage-associated genes *Gata4*, *Gata6*, *Sox7*, and *Sox17*[19]. In the Epi-like cells, we

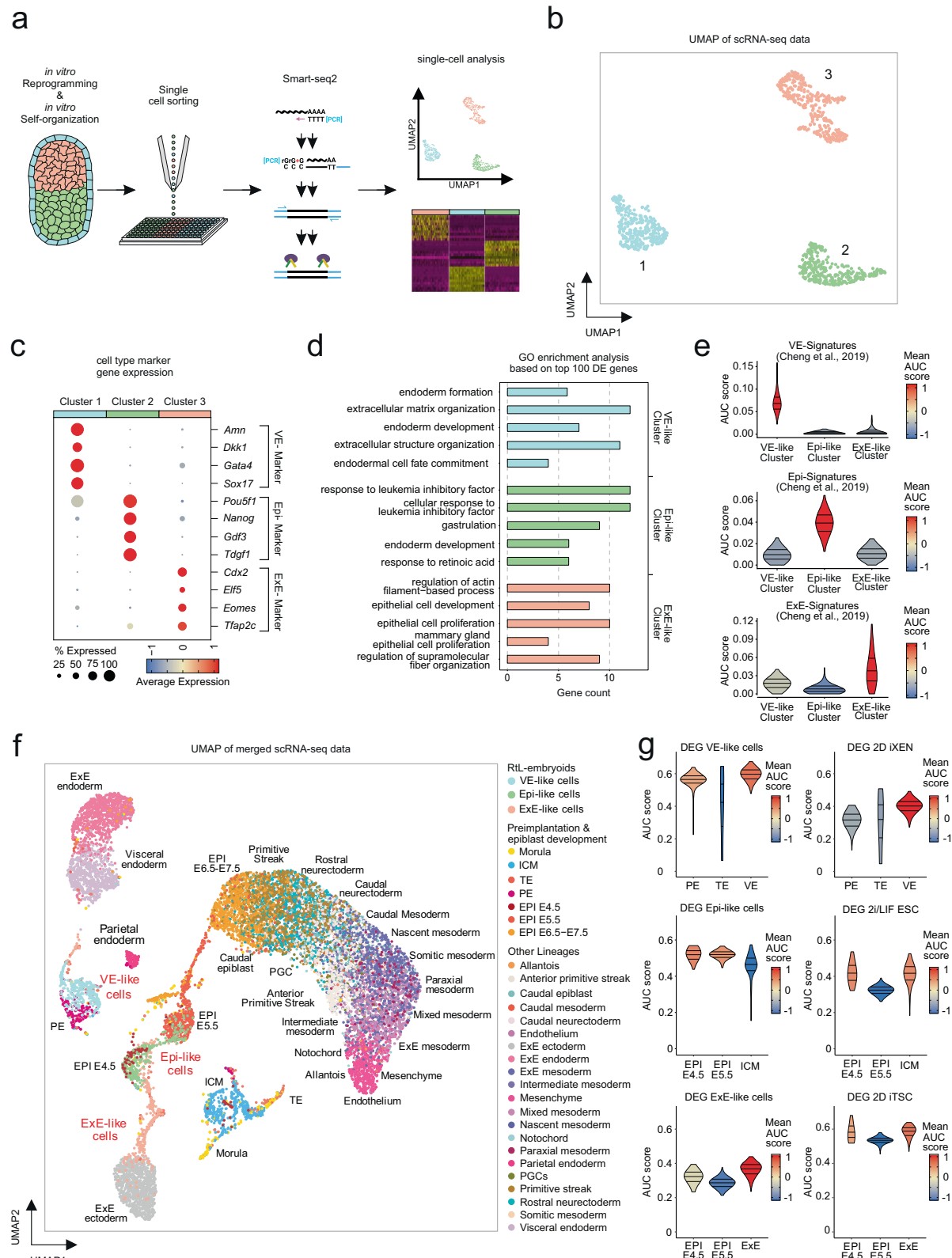

found an enrichment of known embryonic stem cell transcription factors (*Pou5f1* and *Sox2*) but the two transcription factors with the strongest enrichment scores were *Tcf3* and *Nac1*, both of which are essential for regulating ESC differentiation and lineage specification of epiblast cells in gastrulating mouse embryos[20,21]. The ExE-like cluster showed strong enrichment of *Cdx2*, encoding for a core transcription factor responsible for trophectoderm

development[22] (Supplementary Fig. 2g). In line with these findings, we found significant enrichment in a gene signature derived from mouse embryos[23], further supporting the identity of VE-, Epi-, and ExE-like cells (Fig. 2e).

To further evaluate the developmental stages reflected by each of the induced tissues, we compared the scRNA-seq data of RtL-embryoids to datasets derived from cells of murine embryos up to

**Fig. 2 Analysis of transcriptional profiles by scRNA-Seq. a** Schematic representation of assay performed for SMART-seq2 analysis. **b** UMAP representation of scRNA-seq results, showing 3 distinct transcriptomic clusters. **c** Dot Plots showing expression of stem-cell-specific marker genes (*Amn*, *Dkk1*, *Gata4*, *Sox17* = VE-like cells; *Pou5f1/Oct4*, *Nanog*, *Gdf3*, *Tdgf1* = Epi-like cells; *Cdx2*, *Elf5*, *Eomes*, *Tfap2c* = ExE-like cells). **d** GO term enrichment analysis of top 100 differentially expressed genes (log-fold change 1.5; FDR *p* < 0.05). Statistical test: GO terms were selected from the top sorting by adjusted P-value (*p* adjusted < 0.05, one-tailed hypergeometric test with Benjamini–Hochberg correction). Bars depict fold enrichment for terms with *p* < 0.05. **e** AUCell-based enrichment scores (AUC scores) showing similarity of gene expression signatures of the three clusters compared to their respective natural counterparts as assessed by Cheng et al.[23]. **f** UMAP of four integrated reference datasets tracking cells during developmental stages from morula to E7.5 gastrulation in murine embryonic and extraembryonic lineages, in comparison to RtL-embryoid VE-, Epi-, and ExE-like cells. **g** AUCell-based enrichment scores comparing gene signatures of 2D mono-culture and 3D co-culture induced ESCs, TSCs, and XEN cells to natural murine embryo cell clusters from **f**. Source data of **d** are provided as a Source Data file.

E7.5[24–28] (Fig. 2f and Supplementary Fig. 2h). Epi-like cells clustered with both, E4.5 and E5.5 EPI cells, indicating a progression in their developmental stage and supporting our observation of rosette-to-lumen formation (Fig. 2f and Supplementary Fig. 2i). Interestingly, VE-like cells displayed high similarity with transcriptional signatures of primitive endoderm cells and did not show indications of progression towards a visceral endoderm-like identity. ExE-like cells showed the largest overlap with transcriptional signatures of cells derived from the extraembryonic ectoderm or the trophectoderm, again indicating a developmental progression within the trophectoderm lineage (Fig. 2f and Supplementary Fig. 2i). The remaining ExE-like cells clustered to EPI cells from murine embryos at E5.5. Since we further detected weak expression of *Pou5f1* and *Nanog* within cells of the ExE-like cluster (Fig. 2c), we hypothesize, that this subpopulation might represent cells with, at this point of time incomplete reprogramming from ESC to iTSC fate, having adopted an ExE-like transcriptional profile, but still demonstrating transcriptional footprints of the starting (ESC) population.

**Comparison 2D mono-culture and 3D co-culture reprogramming.** To assess whether these 3D co-culture induced VE-like and ExE-like cells differ from their 2D mono-culture induced equivalents, we performed ESC-to-iXEN and ESC-to-iTSC conversions in individual 2D cultures in either serum-based XEN cell- or TSC medium supplemented with FGF4 and Heparin, respectively. After three days of DOX mediated transgene expression and reprogramming, and an additional 24 h w/o DOX cells were harvested and subjected to the same experimental procedure as cells obtained from reprogramming in 3D co-culture (Fig. 2a). ScRNA-seq revealed that Epi-like cells obtained from 3D co-culture in RtL-embryoids and their equivalent cultured in 2D mono-culture in 2i/LIF supplemented ESC medium differed significantly in transcriptional profiles, supporting the presumed progression in developmental stages (Supplementary Fig. 2j). XEN- and TSC-like cells revealed close, but not overlapping clustering of transcriptional profiles with their 3D equivalents. Furthermore, we identified the DEG between the 3D co-culture induced cells and their 2D mono-culture induced equivalents. We compared them with the datasets derived from cells of murine embryos shown in (Fig. 2f) via signature enrichment analysis. We could show that the VE-like cells enriched higher in the VE and PE signatures of murine embryos compared to XEN cells induced in 2D mono-culture (Fig. 2g). The same was true for Epi-like cells, which showed an enrichment in the signatures of Epi E4.5 and E5.5. The 2i/LIF ESCs signature enriched in the EPI E4.5 and ICM and was overall lower compared to the Epi-like cells. The ExE-like cells showed a clear enrichment in the ExE signature of murine embryos, but also displayed weak enrichment in the signature of E4.5 epiblast cells. 2D iTSC signature exhibited a higher enrichment in the murine ExE signature but also a higher enrichment in the Epi E4.5 cluster, compared to their 3D induced equivalents (Fig. 2g). Collectively, scRNA-seq

analysis strongly supported the transcriptional reprogramming of ESCs into distinct cells within the VE-, Epi-, and ExE-like compartmented structures. Additionally, scRNA-seq analysis revealed close, but not overlapping transcriptional profiles of 2D mono- and 3D co-culture induced XEN- and TSC-like cells, suggesting that the cells in 3D cultures not only reprogram but also differentiate further presumably along the induced cell fate. Of note, we were able to rederive ESC and TSCs by cultivating RtL-embryoids in 2D tissue culture dish supplemented with 2i/LIF (for ESC rederivation) and FGF4/Heparin (for TSC rederivation) respectively. These colonies displayed typical dome shaped ESC or flat-TSC colony morphologies, proliferated, could be passaged >20 times and stained positive for ESC markers NANOG and OCT4 or TSC markers CDX2 and EOMES, respectively (Supplementary Fig. 2k–n). We were not able to rederive XEN cells from RtL-embryoids by published XEN cell derivation protocols[12,29]. Whether this reflects a technical problem on our side or an inherent problem of the RtL-embryoid remains to be elucidated.

**The VE-like compartment initiates DVE/AVE specification.** In RtL-embryoids iXEN cells reprogrammed from iGATA6 ESCs assemble into a VE-like sphere, enclosing the two inner compartments (Fig. 1b, c and Supplementary Fig. 1c). Among the top 30 DEGs of the VE-like cluster, specific VE marker genes such as *Amn*, *Sox17* and *Cubn* were identified[30–33] (Supplementary Fig. 2f). Next, we assessed the relationship of this VE-like cluster with published datasets describing VE-lineages in murine embryos. The VE-like cluster shows two transcriptionally diverging subclusters showing similarities to either an ExVE or an EmVE in marker gene expression profiles previously introduced[23] (Fig. 3a, Supplementary Fig. 3a). We tested for enrichment of a previously described EmVE-signature[23] in our VE-like subclusters, clearly showing that this signature is enriched in cluster 2, which we then termed EmVE-like cluster (Fig. 3b). We then aimed to characterize whether RtL-embryoids show further signs of spatio-temporal organization. Here, the formation of distal/anterior visceral endoderm (DVE/AVE) and consequently anterior-posterior axis formation was analyzed. This specification is initiated with the generation of a specialized subpopulation of cells within the distal VE (DVE), expressing the T-box transcription factor Eomesodermin (*Eomes*). This transcription factor plays an instructive role in activating and recruiting further AVE factors, including, *Otx2*, *Lhx1*, and *Hhex*, as well as Wnt and Nodal signaling antagonists *Lefty1*, *Dkk1*, and *Cerl*[34–37] (Fig. 3c). The cells representing DVE then migrate towards one side of the Epi/ExE junction, where they localize next to a second emerging signaling center, the anterior VE (AVE). Together DVE and AVE continue to secrete Wnt and Nodal inhibitory signals, thereby establishing an anterior-posterior axis within the developing embryo. Assessing expression levels of AVE marker genes within our dataset revealed that the majority of AVE marker genes, such as *Lefty1*, *Nodal*, and *Fgf5*, were higher expressed in cells of the EmVE-like cluster (Fig. 3d)[38]. We further corroborated this

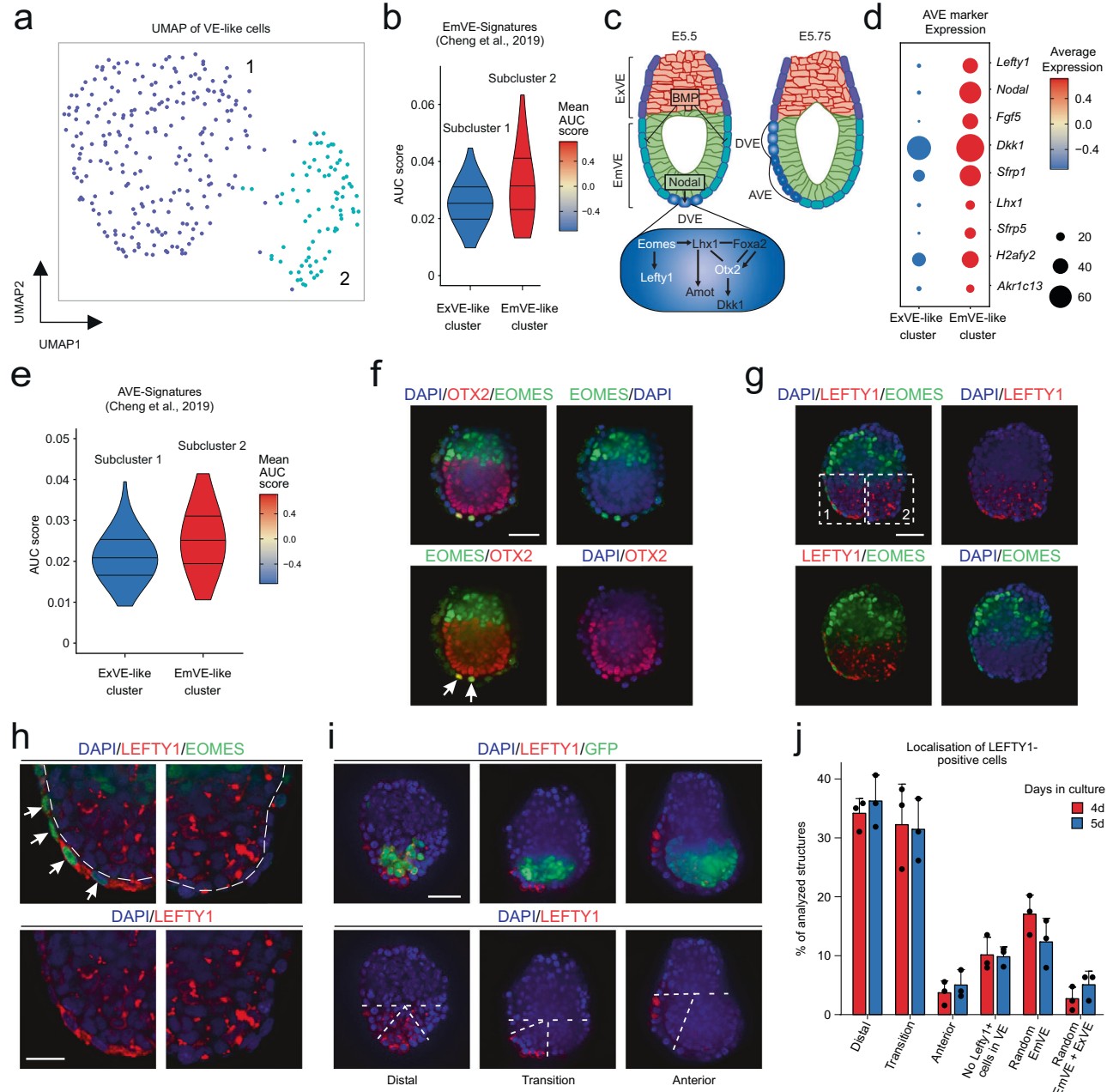

**Fig. 3 Formation of DVE/AVE-like population in VE-like compartment. a** UMAP representation of VE-like cluster revealing two subclusters diverging in their transcriptional profile. **b** Comparison of EmVE-like cluster with published scRNA-Seq dataset for EmVE of natural murine embryos[23] revealing high AUCell-based enrichment score and mean expression levels of EmVE genes within EmVE-like cluster. **c** Schematic representation of DVE and AVE formation in murine embryos between E5.5 and E5.75. **d** Analysis of AVE marker expression in VE-like subclusters displays higher expression of AVE marker gens inside the EmVE-like cell cluster. **e** Comparison of published AVE-signatures[23] showing high AUCell-based enrichment score and high mean expression for cells of EmVE-like cluster, in contrast to a lower AUC score and mean expression levels in cells of ExVE-like cluster. **f** Expression of DVE/AVE markers EOMES and OTX2 was detected to colocalize in cells of the EmVE-compartment in RtL-embryoids, indicated by white arrows. OTX2, red; EOMES, green; DAPI, blue. Scale bar = 50 μm. **g** DVE/AVE marker LEFTY1 was found colocalizing with EOMES + cells, which was detected to be restricted to one side of the EmVE-like compartment in RtL-embryoids (white squares are magnified in **h**. Scale bar = 25 μm. **h** Magnification of the EmVE-like compartment as indicated by the white, dotted squares in **g**. LEFTY1, red; EOMES, green; DAPI, blue. Scale bar = 20 μm. White arrows indicate EOMES + / LEFTY1 + cells. **i** LEFTY1 + cells were detected in distal, transition or anterior position in RtL-embryoids. LEFTY1, red; GFP, green; DAPI, blue. Scale bar = 50 μm. **j** Quantification of LEFTY1 + cell location in RtL-embryoids at day 4 (4d) and day 5 (5d) into the protocol. The majority of RtL-embryoids displayed LEFTY1 + cells in either distal or transition positions, with few RtL-embryoids showing localization of LEFTY1 + in an anterior position, relative to the Epi-like compartments, both 4d (red bars) and 5d (blue bars) into the protocol. Total *n* = 375 for each, 4d and 5d conditions. Data are presented as mean values ± SD. Source data of **j** are provided as a Source Data file. Experiments were repeated independently at least three times with similar results (**f**-**h**).

model by applying enrichment analysis of an AVE signature[23] illustrating that again the EmVE-like cluster showed stronger enrichment for this signature than the ExVE-like cluster (Fig. 3e). As AVE formation initiates anterior-posterior axis development in a spatially confined manner, we then performed immunohistochemistry staining against EOMES, OTX2, and LEFTY1 to visualize their localization (Fig. 3f, g). We detected co-expression of EOMES and OTX2 in cells of the EmVE, which could also be correlated with LEFTY1 expression in these EOMES + / OTX2 + cells (Fig. 3f, g, h). Assessment of localization of LEFTY1 + cells, revealed that the majority of RtL-embryoids showed LEFTY1 + cells in either a distal or transition position, while LEFTY1 + cells were only rarely detected at an anterior position within the EmVE (Fig. 3i, j). Of note, the localization of LEFTY1 + cells did not significantly change between Day 4 and Day 5, indicating that, while in RtL-embryoids a DVE/AVE-like cell population was induced, these cells seem to display a restricted/delayed migratory potential. Interestingly, we observed a failure in restriction of LEFTY1 expression to the DVE in aggregates that displayed weak contribution of ExE-like tissue, most likely due to weak inhibitory BMP signaling from the ExE-like compartment that would be necessary to restrict DVE specification to the distal tip of the VE (Fig. 3c and Supplementary Fig. 3b). Taken together, co-expression of EOMES, OTX2 and LEFTY1, as well as AVE signature enrichment, strongly support the notion that RtL-embryoids initiate DVE/AVE formation. The formation of the anterior-posterior axis seems restricted and further developmental steps might require adjustment in the culture conditions or the overall setup of the RtL-embryoid generation.

**Core- to primed-pluripotency progression in Epi-like cells.** Next, we analyzed the single-cell transcriptomes of the ESC-derived Epi-like cluster in more detail, which displayed a tri-partite character of transcriptionally diverging cell clusters (Fig. 4a). Among the top 30 DEGs characterizing the Epi-like cluster, we identified *Tdgf1* (alias: *Cripto*), *Gdf3*, *Nanog*, and *Pou5f1* (alias: *Oct3/4*), which are bona fide epiblast marker genes[39–45] (Supplementary Fig. 2f). As the VE-like compartment displayed signs of induction of an AVE, we then assessed gene expression for markers of anterior, transition and posterior-epiblast states within the Epi-like cluster[23]. Cells of the Epi-like cluster displayed high signature enrichments with the published anterior-epiblast signature[23] (Fig. 4b). This observation was further supported by the presence of anterior-, low levels of transi-tion- and an absence of posterior-marker gene expression (Fig. 4c). Additionally, GO terms enriched in Epi-like cell marker genes included 'embryonic pattern specification', 'embryonic axis specification', and 'anterior-posterior pattern specification'. Together, these results and previously discussed DVE/AVE for-mation within the VE-like compartment suggest that RtL-embryoids are either situated at the onset of or fail to complete anterior-posterior axis formation.

As RtL-embryoids were shown to undergo rosette formation and lumenogenesis, we next assessed if Epi-like cells progress from naïve- to primed- pluripotency. As described by Neagu et al.[46] progression from a naïve-pluripotency state (KLF4+, NANOG+, ESRRB+, OTX2+, POU3F1− (alias: OCT6)) to a primed-pluripotency state (KLF4−, NANOG−, ESRRB−, OTX2+, OCT6+) occurs between E5.0 and E5.5[46]. In addition to the switch in transcription factor expression during the transition from rosette to lumen stage, this development is accompanied by pulses of phosphorylated ERK (pERK) in epiblast cells, that increase in frequency during progression from rosette to lumen stage[46].

In RtL-embryoids a similar switch of transcription factor expression was observed, as we detected downregulation of KLF4 and ESRRB during rosette to lumen-stage progression, while OTX2 was detected to be expressed at both stages (Fig. 4d–f). Co-Expression of OCT6 and OTX2 was observed in cells at lumen stage, suggesting that cells within the Epi-like compartment of RtL-embryoids indeed progress from a naïve- to primed-pluripotency state (Fig. 4g). However, the naïve-pluripotency marker NANOG was found to be expressed in Epi-like cells throughout rosette and lumen stage, suggesting that in RtL-embryoids progression from naïve- to primed-pluripotency might be delayed or slowed down (Fig. 1i and Fig. 4h). Contributing to this, single pERK+ Epi-like cells were rarely, but exclusively detected in RtL-embryoids at lumen stage, further supporting the notion, that the developmental potential of RtL-embryoids is restricted or delayed in its progression to a completely primed-pluripotency state (Fig. 4i). Of note, KLF4 expression was also detected in cells of both ExE- and VE-like compartments (Fig. 4d). Assessment of marker gene expression of core-, naïve-, and primed- pluripotency factors[27,47] by scRNA-seq confirmed these observations and allowed for the identification of two of the three diverging cluster of Epi-like cells, representing cells in either naïve- or primed-pluripotent states (Fig. 4a, j). While core-pluripotency factors (*Fgf4*, *Utf1*, *Gdf3*, *Tdgf1*, *Pou5f1*, and *Sox2*) were expressed throughout Epi-like subclusters, subcluster 1 displayed the highest expression levels of naïve-pluripotency factors (*Nanog*, *Tbx3*, *Tfcp2l1*, *Fbxo15*, *Esrrb*, *Zfp42*, *Dppa3*, *Klf2*, *Klf4*, and *Klf5*) and lowest expression levels for primed-pluripotency factors. In contrast to this, Epi-like subcluster 2 was characterized by lower expression levels of naïve-pluripotency factors and high expression levels of primed-pluripotency factors (*Nodal*, *Lef1*, *Fgf5*, *Pou3f1*, and *Otx2*) (Fig. 4j). Of note, expression of primed-pluripotency factors *Foxa2*, *Cer1* and *T* was only detected in single cells of the Epi-like cluster, providing further evidence for the presumed delay in progression from naïve- to primed-pluripotency. Assessment of pluripotency factor expression among all three compartments of RtL-embryoids revealed that both, VE- as well as ExE-like cells, predominantly downregulated or silenced expression of pluripotency associated factors (Supplementary Fig. 4a).

**RtL-embryoids show signs of PGC specification.** As RtL-embryoids resemble natural murine embryos at E4.5–E5.5 we analyzed our scRNA-seq dataset for expression signatures indi-cative for PGCs. Specification of PGCs is initiated by BMP4 and BMP8b secretion from cells of the ExE directly adjacent to the epiblast[48–50]. Additionally, PGC specification further relies on BMP2 signaling from the VE[51]. Together, these signals induce expression of *Prdm1* (alias: *Blimp1*) and *Prdm14*, which regulate expression of germ cell development-specific genes *Tfap2c*, *Dppa3* (alias: *Stella*), *Nanos3*, and *Kit*[52–56] (Supplementary Fig. 4b). We detected *Bmp4* and *Bmp8b* expression within a subset of cells of the ExE-like cluster, while *Bmp2* was almost exclusively found to be expressed in the VE-like cluster (Supplementary Fig. 4c). Downstream target genes like *Nanos3*, *Kit*, *Stella*, and *Tfap2c* were found to be expressed within a subpopulation of the Epi-like cluster, with expression of *Nanos3* being highly restricted to this subcluster (Supplementary Fig. 4d). These findings strongly suggest that the spatial organization of RtL-embryoids reflects proper embryonic architecture resulting in BMP-producing cells in their correct compartments and allows for identification of the third Epi-like subcluster as a PGC-like cell population (Fig. 4j).

**Bipartite character of the ExE-like compartment.** Next, we aimed for the characterization of the ExE-like cluster of RtL-

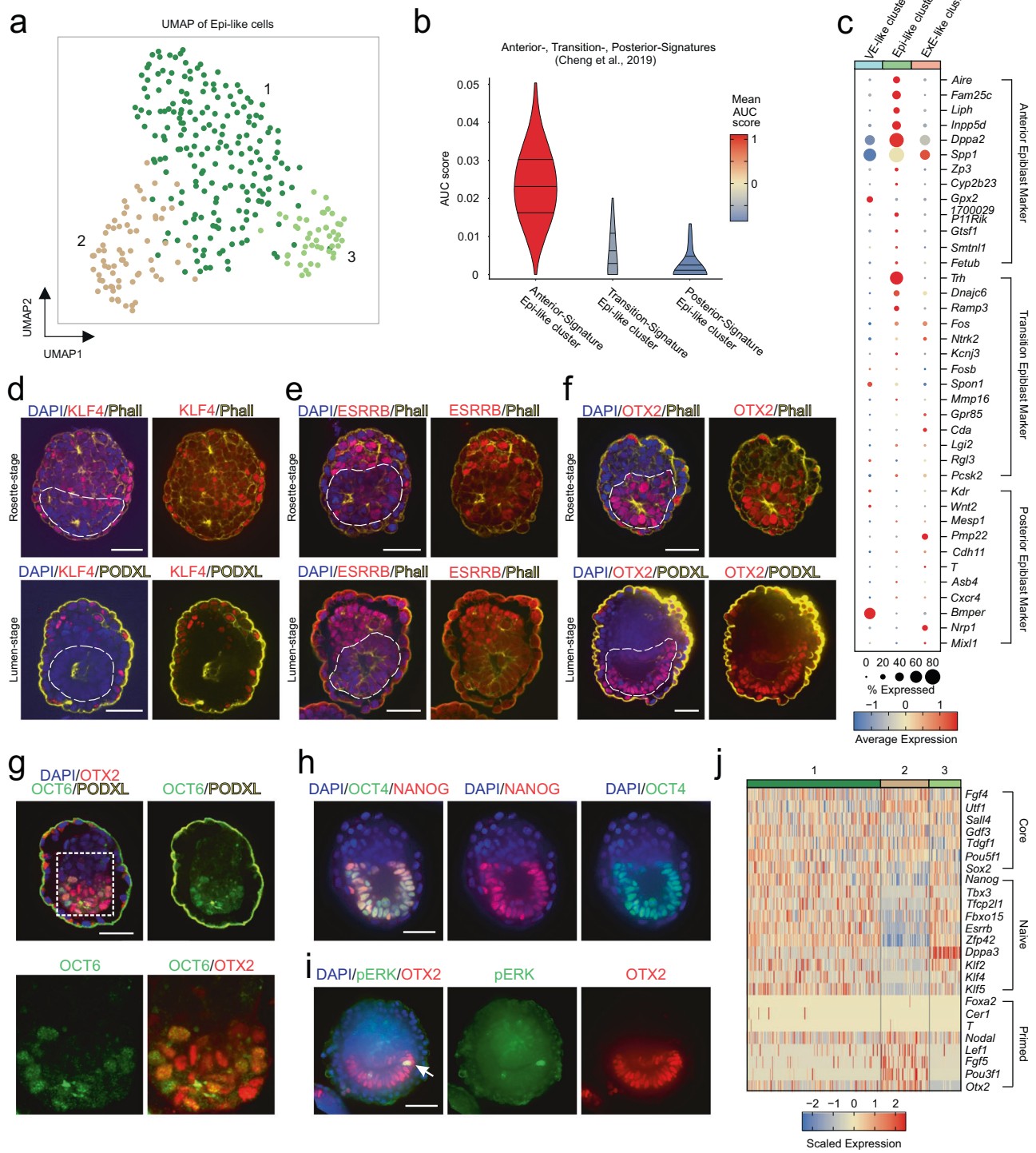

embryoids. Interestingly, while the ExE-cluster displayed an overall ExE-like transcriptional profile, two transcriptionally diverging subpopulations were detected (Fig. 5a and Supplementary Fig. 5a). Ubiquitous trophoblast-/placenta-fate marker genes like *Id2*[57] and *Igf2*[58] were found to be expressed throughout the ExE-like cluster, while expression of key trophoblast marker genes *Eomes*, *Tfap2c*, *Bmp4*, *Bmp8b*, *Hand1*, *Plet1*, and *Elf5*, was predominantly found in cells of ExE-like Subcluster 1 (Fig. 5b). To further predict the subclusters' respective biological functions, we examined the DE genes of diverging ExE-Subcluster 1 and 2 by GO Term analysis of respective cluster marker genes. For ExE-like Subcluster 1, we obtained the highest enrichment scores for

'ribonucleoprotein complex biogenesis', 'placenta development', 'ribosome biogenesis', 'embryonic placenta development', 'embryonic placenta morphogenesis', and 'reproductive system development'. In contrast, ExE-like Subcluster 2 showed the highest enrichment scores for 'negative regulation of neurogenesis', 'negative regulation of cell development', 'negative regulation of nervous system development', 'ossification', and 'regulation of actin-filament-based process' (Fig. 5c). These results suggest diverging biological functions of the ExE-like subclusters.

Such diverging populations of ExE cells have been proposed previously in early murine embryos, in which populations of ExE lineages gradually diversify according to their position within the

**Fig. 4 Progression from naïve- to primed-pluripotency in EPI-like compartment. a** The Epi-like compartment of RtL-embryoids shows three transcriptionally diverging subpopulations, assessed in UMAP representation. **b** Comparison to Anterior-, Transition-, and Posterior-Epiblast signatures published by Cheng et al.[23] revealed highest transcriptional similarity with anterior-epiblast cells. **c** Marker gene expression for Anterior-, Transition-, and Posterior-marker genes. **d, f** IF staining against pluripotency markers during progression from rosette- to lumen stage. The Epi-like compartment of RtL-embryoids (indicated by dotted lines) displays downregulation of naïve-pluripotency markers KLF4 and ESRRB during progression from rosette to lumen stage. Expression of OTX2 was detected at both, rosette and lumen stage. Scale bars = 50 μm. KLF4/ESRRB/OTX2, red; Phalloidin (Phall)/ PODXL, yellow; DAPI, blue. **g** Primed-pluripotency marker OCT6 was detected to be weakly expressed in some OTX2 + cells at lumen stage; Lower Panel shows magnification of area indicated in panel above. Scale bars = 50 μm. OTX2, red; OCT6, green; PODXL, yellow; DAPI, blue. **h** Expression of OCT4 and NANOG was detected in RtL-embryoids throughout the culture period. Scale bars = 50 μm. OCT4, green; NANOG, red; DAPI, blue. **i** pERK pulses were detected in single Epi-like cells of RtL-embryoids at lumen stage, in addition to a diffuse and weak pERK activity in the ExE-like compartment. Scale bars = 50 μm. OTX2, red; pERK, green; DAPI, blue. White arrow indicates pERK+ Epi-like cell. **j** Single-cell heatmap of core-, naïve-, and primed-pluripotency markers among cells of the Epi-like Subclusters, revealing predominantly naïve-pluripotency factor expression in subcluster 1, while subcluster 2 displayed downregulation of naïve- and upregulation of primed-pluripotency factor in subcluster 2. Subcluster 3 displayed a PGC-like character. Experiments were repeated independently at least three times with similar results (**d–i**).

ExE[59]. The ExE cells directly adjacent to the epiblast are therefore referred to as proximal (Pr)ExE, while ExE cells lining the ectoplacental cone are referred to as distal (Di)ExE[59]. To investigate this subdivision of the ExE-like cluster, we analyzed the expression patterns of potential signaling molecules of the Epi-like compartment that might influence the development in the ExE-like compartments. As FGF signaling is known to be necessary for the proliferation of TSCs and FGF ligand and receptor expression is highly tissue-specific within natural developing embryos[2,60–62], we evaluated their expression among RtL-embryoids (Fig. 5d and Supplementary Fig. 5b). The expression of *Fgf4* was found exclusively within the Epi-like cluster, while the expression of its receptor *Fgfr2* was predominantly found in Subcluster 1 of the ExE-like cluster. TSC genes downstream of FGFR2, like *Cdx2*, *Eomes*, *Esrrb* and *Elf5* were also found to be expressed within Subcluster 1, with expression of *Eomes* and *Elf5* being predominantly restricted to this subcluster, thereby again mirroring the situation in murine embryonic development[63] (Fig. 5e and Supplementary Fig. 5b). In natural embryogenesis, *Fgfr2* and *Eomes* are expressed in the ExE cells adjacent to the epiblast and downregulated towards the ectoplacental cone[63,64]. Thus, we hypothesize that ExE-Subcluster 1 might be the stem cell niche within the ExE-like compartment adjacent to the Epi-like compartment (PrExE-like cluster), while ExE-Subcluster 2 represents more differentiated cell fates, resembling the DiExE.

Next, we analyzed the expression of proliferation markers and cell cycle stage distributions among cells of the ExE-like Subclusters. Proliferation markers *Pcna*, *Top2a*, *Mcm6*, and *Mki67* were highly expressed within the majority of cells of ExE-like Subcluster 1, compared to lower expression levels in smaller proportions of cells of the ExE-like Subcluster 2 (Supplementary Fig. 5c). Highlighting the stem cell character of ExE-Subcluster 1, the comparison with iTSCs cultured under FGF4/Heparin supplementation, revealed highly similar expression patterns for each of the proliferation markers (Supplementary Fig. 5c). Interestingly, ExE-Subcluster 2 displayed a broader distribution of proliferation marker gene expression levels, ranging from cells in which expression of proliferation markers *Pcna*, *Top2a*, *Mcm6*, and *Mki67* was completely absent, to cells showing identical expression levels as ExE-like Subcluster 1 (Supplementary Fig. 5c). Next, we assessed cell cycle stage distributions[65]. We found the majority of cells of ExE-Subcluster 1 to be in the S (35.07%) and G2M (57.46%) phases, with few cells in the G1 phase (6.71%), as observed in proliferating trophoblast stem cells[66] (Fig. 5h). In contrast, the majority of cells within ExE-Subcluster 2 were found to be in the G1 (46.04%) phase and fewer cells in the S (25.24%) and G2M (28.71%) phases, indicative of decreased self-renewal capacity and increased differentiation

state (Fig. 5h). Supporting these observations, we found *Id2*, encoding for an important regulator of placental differentiation, to be expressed in both ExE-like Subclusters, albeit at higher levels in the majority of cells in ExE-Subcluster 1, as is the case for proliferative TSCs, and lower levels in ExE-Subcluster 2, as expected for cells undergoing differentiation into lineage-specific trophoblast subtypes[57] (Fig. 5b). To further analyze this bipartite character of the ExE-like compartment, we performed IF staining against pERK, as the ExE has been shown to exhibit a specific spatial and temporal pattern of pERK signaling during mouse embryogenesis[67]. At E5.5 pERK can be detected throughout the ExE and the distally emerging ectoplacental cone[67]. At this stage pERK signaling is mediated either FGFR dependent (in the PrExE lining the epiblast) or FGFR independent (in the DiExE and the emerging ectoplacental cone)[67]. At E6.0–E7.5 strongest pERK signaling is restricted to a narrow band within the PrExE lining the epiblast (FGFR dependent) and the ectoplacental cone (FGFR independent) emerging from the DiExE[67]. In RtL-embryoids, IF staining against pERK supported the presumed gradient of differentiation, as the strongest pERK activity could be observed at the most distal region of the ExE-like compartment, reminiscent of FGFR independent pERK signaling in the ectoplacental cone emerging from the DiExE at E6.0[67] (Fig. 5g). Additionally, we detected pERK signaling in ExE-like cells lining the Epi-like compartment, similar to FGFR-dependent pERK signaling in the PrExE[67] (Fig. 5g). In general, MAPK phosphorylation within the ExE-like compartment was found in either a diffuse distribution spanning the ExE-like compartment as a whole (Fig. 4i) or forming a pattern with the strongest activity of pERK at the most distal position within the ExE-like compartment and a weaker activity in ExE-like cells lining the Epi-like compartment (Fig. 5g). Therefore, spatial and temporal patterns of pERK signaling in the ExE-like compartment are similar to pERK signaling patterns in murine embryos between E5.5 and E6.5[67]. To further characterize the two ExE-like subclusters, we compared their transcriptional profiles with 2D mono-culture induced iTSC, that were derived under FGF4/Heparin supplementation. The comparison revealed close transcriptional clustering of iTSCs obtained from FGF4/Heparin supplemented 2D reprogramming with ExE-like Subcluster 1, both of which displayed high expression of *Fgfr2* (Fig. 5f). Additionally, 2D mono-cultured reprogrammed iTSCs and cells of ExE-like Subcluster 1 displayed similar distributions of cell cycle phases, highlighting the stem cell character of this subcluster (Fig. 5h and Supplementary Fig. 5c).

**Predicted ligand-receptor interactions in RtL-embryoids.** We next set out to identify the signaling interactions between the three major compartments by more unbiased ligand-to-target

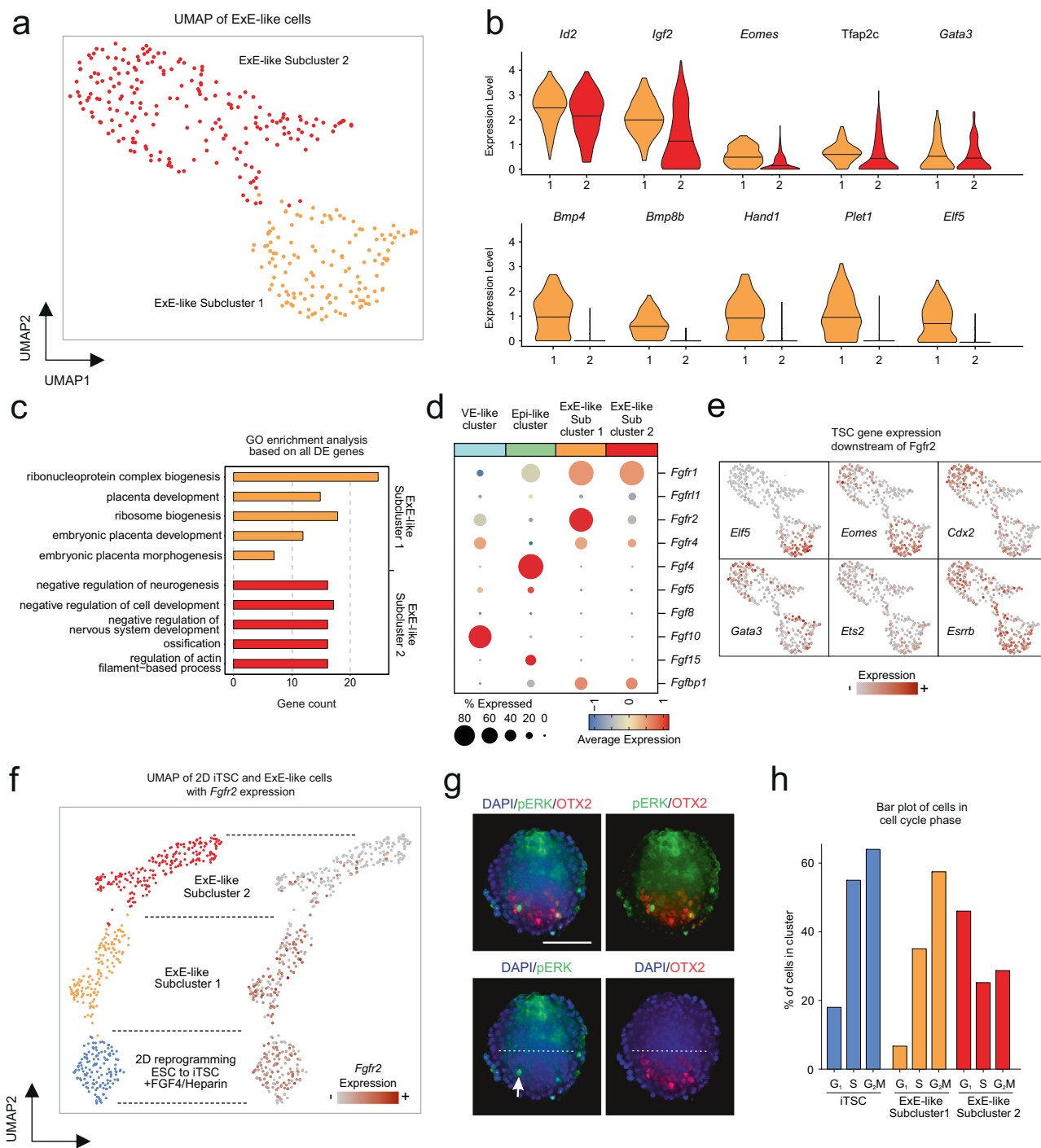

**Fig. 5 Bipartite transcriptional character of cells comprising the ExE-like cluster. a** UMAP representation reveals two transcriptionally diverging subclusters for the ExE-like cluster. **b** Violinplots show expression of trophoblast marker genes. The plots display cells with an expression >0 (line represent mean of cells). **c** GO Term analysis based on all variable genes of each of the ExE-like subclusters, revealed diverging biological functions of the two subcluster. **d** Dotplot displaying FGF-receptor and ligand expressing cells throughout VE-, Epi-, and ExE-like (Sub)clusters. **e** Featureplots depicting expression of FGF4 downstream signaling targets within the ExE-like cluster. **f** UMAP representation including transcriptional profiles of iTSCs obtained from reprogramming in FGF4/Heparin supplemented 2D mono-culture, shows close clustering of iTSCs and ExE-like Subcluster 1, both of which display high expression of the FGF4 receptor *Fgfr2*. **g** IF staining against pERK showed most intense pERK activity in the distal part of the ExE-like compartment. Scale bar = 100 μm; White arrow indicates pERK+ cell in Epi-like compartment; Dotted lines represents border of Epi- and ExE-like compartments. OTX2, red; pERK, green; DAPI, blue. **h** Bar plots displaying distribution of cell cycle phases in ExE-like Subcluster 1, 2 and iTSCs, as assessed by Nestorowa et al.[65], confirming stem cell characteristics of ExE-like Subcluster 1 and differentiating character of ExE-like Subcluster 2. Experiments were repeated independently at least three times with similar results (**g**).

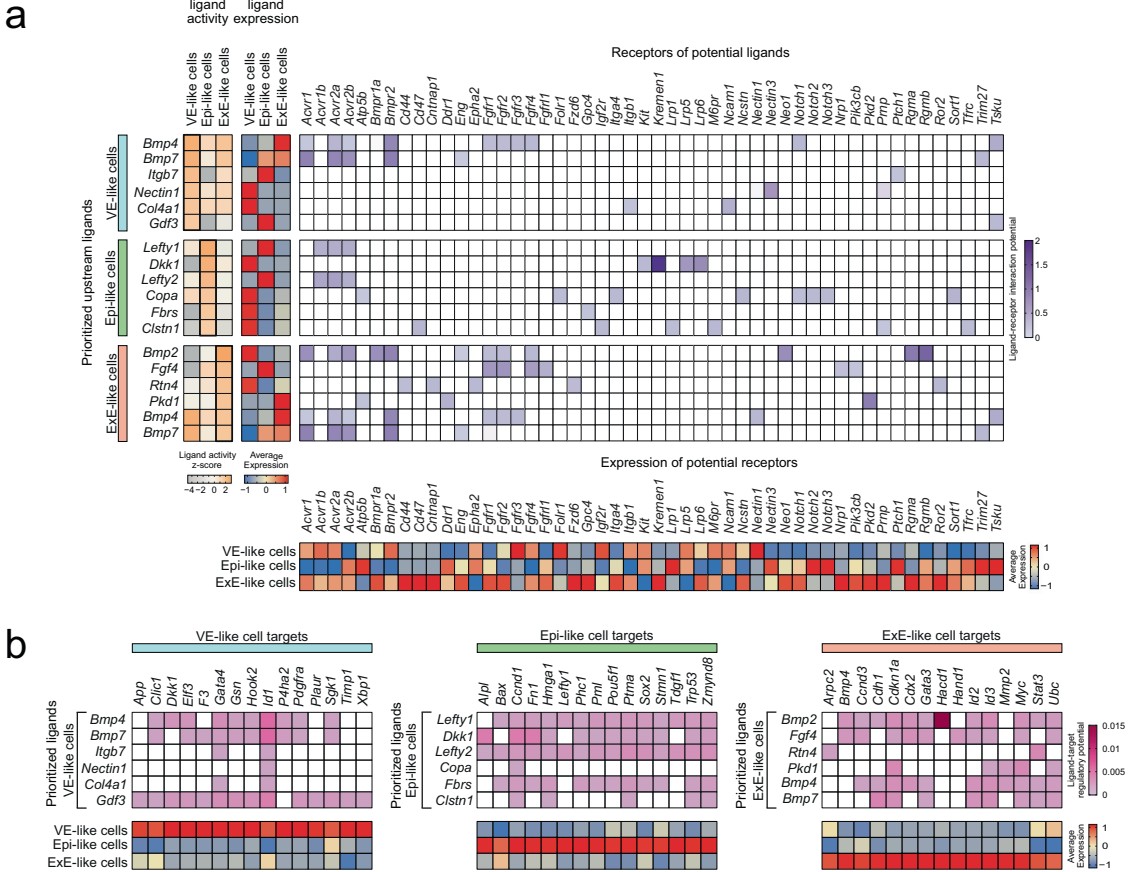

**Fig. 6 Ligand-to-target interaction landscape of major compartments in RtL-embryoids. a** Prioritized upstream ligands from all embryo-like compartments (left panel) based on the interaction with all other cells and their average expression (middle panel); In the right panel are potential receptors expressed by the corresponding compartment and (bottom) their average expression. **b** In the top panel are potential target genes of the prioritized upstream ligands and on the bottom panel their average expression.

signaling analysis. To that end, we used the NicheNet algorithm that predicts ligand-receptor interactions and the subsequent gene expression by combining annotated single-cell RNA-seq data with existing knowledge on signaling and gene regulation[68] (Fig. 6a). The top six ligands predicted to exert the highest activity within the VE-like compartment (receiver compartment) were bone morphogenetic protein 4 (BMP4), bone morphogenetic protein 7 (BMP7), integrin subunit beta 7 (ITGB7), nectin cell adhesion molecule 1, (NECTIN1), collagen, type IV, alpha 1 (COL4A1), and growth differentiation factor 3 (GDF3) (Fig. 6a). VE-like cells express genes encoding for important BMP4 and BMP7 receptors like Activin A receptor 1 (*Acvr1*), Activin A receptor 2a (*Acvr2a*), and bone morphogenetic protein receptor type II (*Bmpr2*). Our analysis also revealed that *Bmp4* and *Bmp7* are mainly expressed and potentially secreted by the ExE-like compartment, which would serve as the sender compartment (Fig. 6a). Considering their known importance for induction and migration of the AVE and induction of PGC specification[69,70], it can be assumed that Smad signaling is present and active in RtL-embryoids. The detection of GDF3 among the top six ligands, showing the highest activity in the VE-like cluster, further supports this notion, as it is known as a key regulatory element controlling AVE formation[40]. Besides, these data strengthen our observation of PGC specification within cells of the epiblast-like cluster, as BMP signaling mediated by ACVR1 (ALK2) in the visceral endoderm is known to be necessary for the generation of PGCs in the mouse embryo[69]. In a second step, we predicted the target genes for the top six ligands in the individual

compartments (Fig. 6b). This underlined the important overlap in the VE-like cells between BMP-induced and potential GDF3-induced target genes like *Dkk1*, *Elf3*, and *Gata4*.

The top 6 predicted ligands for the Epi-like compartment were left-right determination factor 1 (LEFTY1), Dickkopf-related protein 1 (DKK1), left-right determination factor 2 (LEFTY2), Coatomer protein complex subunit alpha (COPA), fibrosin (FBRS), Calsyntenin1 (CLSTN1) (Fig. 6a). LEFTY1 and LEFTY2 are known to inhibit Nodal activity in the Epiblast by indirect interaction with the Nodal receptors ACVR1B, ACVR2B, and antagonizing EGF-CFC co-receptors[71,72]. In our dataset, we observed the highest ligand-receptor interaction potential of LEFTY1 and LEFTY2 with ACVR1B, ACVR2A, and ACVR2B of which *Acvr2b* showed high gene expression within cells of the EPI-like cluster (Fig. 6a). Therefore, it can be assumed that LEFTY1/2 signaling within RtL-embryoids mirrors the situation in murine embryos. DKK1 has been shown to inhibit Wnt/beta-catenin signaling by binding to and antagonizing LRP5/6, presumably by functionally cooperating with high-affinity DKK1 receptors KREMEN1 and KREMEN2[73]. Furthermore, DKK1 expression has been demonstrated to be initiated in the mature primitive endoderm starting around E4.5[46,74] and was proposed to indicate the completion of PrE maturation, repressing WNT signaling and allowing for *Otx2* induction, rosette formation, and pluripotency progression[46]. Within RtL-embryoids we were able to identify DKK1 (originating from the VE-like cluster) as the second most active ligand in cells of the Epi-like cluster, showing high interaction potential with receptors

LRP5/6 and very high interaction potential with KREMEN1. *Lrp5* was found to be expressed in cells of the EPI-like cluster, while expression of the gene *Kremen1*, encoding for its receptor, could be detected as well, albeit at low rates. Together these results indicate that the DKK1-mediated Wnt/beta-catenin inhibiting cascade is present in RtL-embryoids as described for murine embryos. Among the predicted target genes controlled by LEFTY1, DKK1, and LEFTY2 are *Pou5f1* and *Sox2*, which are critically involved in self-renewal, or pluripotency, of ICM and epiblast (Fig. 6b). The ligands showing the highest activity scores within the ExE-like cluster included BMP2, BMP4, and BMP7. BMP2 and BMP4 have been shown to play essential roles in placental developmental[75], while BMP7 functions predominantly as a heterodimer with BMP2 or BMP4 during mammalian embryogenesis[76]. In RtL-embryoids these signaling pathways seem to be present as well. Additionally, the gene encoding for their receptor, *Bmpr1a* is highly expressed in cells of the ExE-like cluster (Fig. 6a)[77]. Furthermore, NicheNet analysis revealed PKD1 as one of the most active ligands acting on the ExE-like cluster, while the gene encoding for its receptor *Pkd2* was found to be highly expressed in cells of the ExE-like cluster as well. The interplay of PKD1 and its receptor PKD2 is required for placental morphogenesis, hence, this signaling-cascade of mammalian embryogenesis seems to be represented in RtL-embryoids as well[78]. Additionally, we show that among the downstream targets of BMP2 and BMP4 in the ExE-like cluster are *Cdx2* and *Gata3* which are both essential for TSCs and placental development (Fig. 6b). Altogether, NicheNet analysis highlighted that the communication between the three main embryonic clusters in RtL-embryoids resembles major pathways previously described to be involved in natural murine embryo development.

## Discussion

In this study, we present a system for the generation of embryo-like structures, termed RtL-embryoids, from a solely ESC starting population, based on combining transcription factor-mediated reprogramming paradigms and 3D co-culture. We demonstrate that the induction of five TSC-fate characteristic transcription factors in one ESC line and one XEN cell fate-related transcription factor in a second ESC line, when cultured with a third, unmodified ESC line, results in the generation of structures resembling natural murine embryos stage E4.5–E5.5. Previous work demonstrated that embryo-like structures can be generated by spontaneous self-assembly of ESCs, TSCs, and XEN cells isolated from blastocysts[8,9,79]. Such approaches require the elaborate culture of three stem cell lines in parallel. Here, we demonstrate that it is possible to use ESCs as the only starting population when transcription factor-mediated reprogramming of such ESCs towards iTSCs and iXEN is combined with simple co-culture conditions of these three ESC-derived cell types. As previously shown, transcription factor-mediated reprogramming can be used to generate iTSC and iXEN cells from mESC[11–14]. Here, we demonstrate that reprogramming towards iTSC and iXEN cell fate in 3D co-culture not only leads to the induction of the respective cell-lineage but also results in compartmented embryo-like structures. Hence, we hypothesize that reprogramming in co-culture in a 3D environment results in crosstalk between the cells undergoing cell-fate conversion, leading to the formation of multicellular, complex embryonic, and extra-embryonic tissues.

In XEN cells, BMP signaling has been shown to induce visceral endoderm differentiation[80]. Within RtL-embryoids, we observed *Bmp4* and *Bmp8b* expression originating from a subpopulation of cells of the ExE-like compartment, comparable to expression patterns in natural murine embryogenesis. Therefore, we hypothesize that overexpression of *Gata6* in mESC combined with BMP4 signaling from the ExE-like compartment induces cellular reprogramming not only towards an iXEN cell identity, but rather also forcing the converting cells towards an induced VE-like fate. Interestingly, we observed not only the induction of a VE-like fate but also the generation of highly specialized VE-lineages, such as the DVE/AVE. This DVE/AVE-like cell population, however, did display limited migratory potential and few aggregates displayed localization in a truly anterior position relative to the Epi-like compartment. Corresponding with this observation we detected enrichment in an anterior-epiblast gene signature within Epi-like cells, but no enrichment in transition or posterior-epiblast gene signatures. Assessment of pluripotency states within Epi-like cells suggests a progression from naïve- to primed-pluripotency, similar to published roadmaps of pluripotency progression during rosette to lumen maturation[46]. However, as we observed a failure in downregulation of naïve-pluripotency marker *Nanog*, and no expression of the primed-pluripotency markers *T*, *Cer1*, and *Foxa2*, we assume that the Epi-like compartment of RtL-embryoids might not progress to a completely primed-pluripotency state.

Additionally, we detected signs of PGC formation within a subset of cells of the Epi-like cluster. PGC specification in mice is initiated through a gradient of BMP2/4 signaling; however, cells of the EPI-like cluster did not show any population overlapping with the transcriptional signatures of PGC obtained from murine embryos (Fig. 2f). Hence, PGC specification observed here must follow a different route of transcriptional induction. It was previously demonstrated that PGC-like cells can be derived from Epi-like cells in response to BMP and WNT exposure[81]. The PGC specification observed here is therefore most likely the result of signaling cascades mediated by BMP4 and BMP8b secretion from the ExE-like compartment adjacent to the Epi-like compartment, instead of the natural specification route[48–51,81]. Another example of instructive interaction between Epi- and ExE-like compartments was observed in the FGF4 mediated proliferation of a stem cell niche within the ExE-like compartment. FGF4 signaling from the epiblast is known to be required for proliferation of the TSC compartment within the ExE[2,60–62]. Again, RtL-embryoids display resemblance to natural murine embryos, as *Fgf4* is expressed in the Epi-like compartment, while its receptor *Fgfr2* and its downstream target genes were found to be expressed within a subpopulation of cells of the ExE-cluster. Supporting this observation, we detected a pattern of pERK signaling in the ExE-like compartment similar to the situation found in the ExE between E5.5 and E6.5[67].

Taken together, these findings indicate that crosstalk between the compartments of RtL-embryoids leads to more complex cell-fate conversions, in addition to the transcription factor-mediated reprogramming into iTSC and iXEN cell fates. Furthermore, the system presented here may provide a tool for the enrichment and isolation of rare stem cell lineages such as DVE/AVE-like and PGC-like cells. The simplified approach using a solely ESC based starting cell population eliminates the need for costly, individual cell culture reagents required for maintenance and proliferation of different stem cell lines (e.g. ESC, TSC, and XEN). In addition, using the 3D mold techniques, >700 correctly assembled RtL-embryoids can be generated in a single 12-well plate. RtL-embryoids develop an early murine embryo architecture of inner-embryonic and extraembryonic compartments and display rosette formation, as well as progression to lumen, in Epi- and ExE-like compartments. As such, they can model the development of epiblast, visceral endoderm and extraembryonic ectoderm of murine embryos between E4.5 and E5.5. Recently, Rossant and Tam[82] proposed a distinction between models that mimic the fundamental parts of the whole conceptus and models that mimic

selective morphogenetic events[82]. Models like gastruloids that recapitulate selective events, would therefore be regarded as non-integrated stem-cell-based embryo models[82]. Models like blastoids, ETX-embryoids, as well as RtL-embryoids presented in this study would therefore be referred to as integrated stem-cell-based embryo models, as they model multiple interacting tissues of the conceptus[82]. In the growing field of stem-cell-based embryo models, several terms like artificial or synthetic are currently used to describe the structures generated. Rossant and Tam[82] noted that such synthetic embryo models would require new genetic switches or synthetic pathways, preprogrammed into the starting cell population, to be regarded synthetic[82]. The RtL-embryoids presented here are based on the use of cells harboring conditional genetic switches allowing for controlled pathway manipulation by transgene expression. So, our RtL-embryoid system can be regarded a synthetic embryo model. Conclusively, RtL-embryoids should be classified as a synthetic, integrated stem-cell-based embryo model that offers an additional tool to study the complex spatio-temporal regulation of gene expression and cellular communication during early murine embryogenesis.

## Methods

The research performed in the course of this study complies with all relevant ethical regulations.

**Stem cell lines**. Kermit ESCs were derived by our group using standard ES derivation protocols[16] from blastocysts of *Oct3/4_GFP* transgenic mice[83]. The ES cell line KNUT1 was derived by our group as described[16]. *Cdx2, Tfap2c, Eomes, Gata3, Ets2* ESCs (5 Factor / 5F-ESCs) were previously generated and characterized by our group[14]. An inducible XEN cell line was created by lentiviral transduction of ESCs using the pCW57.1_Gata6 plasmid, a gift from Constance Ciaudo (Addgene plasmid #73537; [http://n2t.net/addgene:73537]). Lentiviral particles were produced in 293 T HEK cells (ATCC order number #CRL-3216; Human embryonic kidney cells, harboring a SV40 large T antigen; Cell line received from Dr. Michael Peitz) by co-transfection of the lentiviral pCW57.1_Gata6 plasmid with VSV-G (pMD2.G, Addgene #12259) and helper plasmid (psPAX2, Addgene #12260) by calcium-phosphate precipitation. Virus-containing supernatant was harvested 48 h and 72 h after transfection, pooled and filtered through 0.4 µm SFCA membranes, and stored at −80 °C. A wild-type ESC line established by our group was cultured under standard ESC conditions with feeders until reaching ~60% confluency. At this point, viral transductions were performed overnight in a six-well dish using 800 µl of standard ESC medium with 200 µl virus-containing supernatant supplemented with 8 µg/ml polybrene (Sigma–Aldrich: TR-1003). The selection of positive clones was performed by supplementing the culture medium with 1 µg/ml puromycin for 3 days. All lentiviral protocols were performed under S2 conditions.

**Cell culture of ES cell lines**. ES cells were cultured on gelatin-coated in ES-Medium consisting of DMEM + GlutaMAX (Gibco: 31966-021) supplemented with 2 mM L-glutamine (Gibco: 25030-024), 50 U/ml penicillin/streptomycin (Gibco: 15140-122), 1× nonessential amino acids (Gibco: 11140-035), 1× essential amino acids (Gibco: 11130-036), 0.1 mM β-Mercaptoethanol (Gibco: 31350-010), 15% FCS (Gibco: 10270-106), LIF (1000 U/ml; Sigma–Aldrich: ESG1107), and 2i (3 µM CHIR-99021, Cayman Chemical: 13122; 1 µM PD0325901, BioGems: 3911091). Induced TS cells reprogrammed from 5 Factor ESCs were cultured in TSC Medium consisting of advanced RPMI 1640 (Thermo Fisher Scientific: 11875-093), 20% FCS, 2 mM L-glutamine, 1 mM sodium pyruvate (Thermo Fisher Scientific: 11360070), 50 U/ml penicillin/streptomycin, and 0.1 mM β-Mercaptoethanol. A mixture consisting of 70% of feeder-MEF conditioned TSC medium and 30% fresh, unconditioned TSC medium was prepared and supplemented with FGF4 (Reliatech: #100-017 L; 25 ng/ml), Heparin (Sigma–Aldrich: #H3149-10KU; 1 µg/ml) for the culture of iTSCs. Induced XEN cells reprogrammed from iGATA6 ESCs were cultured on gelatin-coated culture dishes in XEN cell culture medium, consisting of RPMI 1640, 15% FCS, 2 mM L-glutamine, and 50 U/ml penicillin/streptomycin.

**3D cell culture and generation of RtL-embryoids**. 3D cell culture dishes were generated using the MicroTissues® 3D Petri Dish® micro-mold spheroids (Sigma–Aldrich: Z764094-6EA) according to manufacturer's protocol with 2% molten cell culture grade Agarose (Sigma–Aldrich: A9539; in sterile saline (0.9% w/v NaCl)). 3D Petri Dish®, holding 256 microwells, were equilibrated in reconstructed embryo medium[9] for 1 h at room temperature in a 12-well plate. The reconstructed embryo medium consists of 39% advanced RPMI 1640 and 39% DMEM (Thermo Fisher Scientific: 11960069) supplemented with 17.5% FCS, 2 mM L-glutamine, 0.1 mM ß-mercaptoethanol, 0.1 mM MEM nonessential amino

acids, 1 mM sodium pyruvate, 1% penicillin-streptomycin[9]. Before seeding, each of the three ESC lines used for the generation of RtL-embryoids was diluted to cell counts resulting in an average-based seeding of 6 Kermit ESCs (or KNUT1 ESCs), 16 5-Factor ESCs and 5 iGATA6 ESCs per microwell of the 3D Petri Dish®. The diluted ESC populations were then pooled, centrifuged, and resuspended in ES-Medium, before seeding on the 3D Petri Dish®. Cells were cultured with ES-Medium without doxycycline for 24 h to allow the cells to form embryoid bodies, before switching the culture medium to reconstructed embryo medium supplemented with 2 µg/ml doxycycline (Sigma–Aldrich: D9891). Aggregates were cultured under this condition for 3 days, inducing transgene expression and reprograming into iTSCs or iXEN cells, leading to self-organization into RtL-embryoids. Generated structures were harvested from their 3D Petri Dish® by placing the agarose pad upside-down into a new 12-well plate filled with 2 ml PBS and centrifugation at 45 × g for 3 min, forcing the aggregates out of their microwells.

**Quantification of correctly compartmented RtL-embryoids**. For the quantification of compartmentalization of the structures Kermit ESC, 5-Factor ESCs, and iGATA6_mCherry ESCs were used for seeding of the starting cell populations, in average ratios per microwell as previously introduced. Quantification of compartmented structures was performed in three independent experiments, assessing a total of 1167 aggregates (day 4) and 778 aggregates (day 5). Structures were regarded correctly compartment if they exhibited formation of clearly separated Epi- and ExE-like compartments that were surrounded by a complete monolayer of VE-like tissue (see Supplementary Fig. 1c). Structures that were comprised of all three cell lineages but failed to completely segregate into an Epi- or ExE-like compartment or displayed an incomplete VE-like tissue were regarded as incorrect compartment (for example see Supplementary Fig. 1e and f).

**Immunofluorescence staining of RtL-embryoids**. The presence of proteins was detected by immunofluorescence staining. Aggregates were generated and harvested as described above, before being fixed using 4% formalin (Sigma: 100496) for 20 min at 4 °C. After washing the aggregates three times with wash buffer (0.1% Tween-20 (AppliChem: A44974) in PBS) permeabilization was performed with 0.5% Triton X-100 (AppliChem: A4975) in PBS for 30 min at room temperature. Aggregates were incubated with primary antibodies in blocking buffer (3% BSA (Sigma–Aldrich: A9647) and 0.3% Triton X-100 in PBS) at 4 °C overnight. Afterward, aggregates were washed three times to remove unbound primary antibody and incubated with Alexa Fluor-conjugated secondary antibodies in blocking buffer again at 4 °C overnight and protected from light. Staining with phalloidin was performed in blocking buffer for 2 h, after incubation with secondary antibodies. Aggregates were washed again three times, resuspended in Roti®-Mount FluorCare DAPI (Roth: HP20.1) to stain cell nuclei, and transferred on Cellview Cell Culture Dishes (Greiner Bio-One: 627861), allowing for stable three-dimensional imaging without disturbing the aggregates structures. Primary antibodies used and dilutions: Goat-polyclonal anti-CDX2 (Santa Cruz: sc-19478; 1:200), Goat-polyclonal anti-GATA4 (Santa Cruz: sc-1237; 1:400), Goat-polyclonal anti-LEFTY1 (R&D Systems: AF746; 1:200), Rabbit-polyclonal anti-p44/42 MAPK (Erk1/2) (Cell signaling: #9101; 1:100), Mouse-polyclonal anti-OCT6 (Absea: 060204E04; 1:10), Mouse-polyclonal anti-ESRRB (Perseus Proteomics:PP-H6705-00; 1:200), Rabbit-polyclonal anti-NANOG (ReproCell: RCAB002P-F; 1:300), Rabbit-polyclonal anti-EOMES (Abcam: ab23345; 1:400), Goat-polyclonal anti-KLF4 (R&D: AF3158; 1:400), Rat-polyclonal anti-PODXL (R&D: MAB1556; 1:300), Goat-polyclonal anti-OTX2 (R&D: AF1979; 1:400), Rabbit-polyclonal anti-GATA3 (Abcam: ab199428; 1:300), Mouse-polyclonal anti-OCT4 (Santa Cruz: sc-5279; 1:300). Probes and dilutions used: Alexa Fluor 488-Phalloidin (Invitrogen: A12379; 5 units/ml), Alexa Fluor 594-Phalloidin (Invitrogen: A12381; 5 units/ml). Secondary antibodies used and dilutions: Donkey polyclonal secondary antibody to Goat IgG-H&L Alexa Fluor 594 (Abcam: ab150132; 1:50), Chicken polyclonal secondary antibody to Goat IgG-H&L Alexa Fluor 647 (Invitrogen: A-21469; 1:500), Goat-polyclonal secondary antibody to Rabbit IgG-H&L Alexa Fluor 594 (Invitrogen: A-11012; 1:500), Goat-polyclonal secondary antibody to Rat IgG-H&L Alexa Fluor 488 (Invitrogen: A-11006; 1:500), Goat-polyclonal secondary antibody to Mouse IgG-H&L (Alexa Fluor 488) (Invitrogen: A-11001; 1:500), Donkey polyclonal secondary antibody to Mouse IgG-H&L (Alexa Fluor 594) (Invitrogen: A-21203; 1:500), Chicken polyclonal secondary antibody to Rat IgG-H&L (Alexa Fluor 647) (Invitrogen: A-21472; 1:500), Goat-polyclonal secondary antibody to Rabbit IgG-H&L (Alexa Fluor 488) (Invitrogen: A-11008; 1:500).

**Derivation of stem cells from RtL-embryoids**. ESC-like and ExE-like stem cells were derived from RtL-embryoids by outgrowth culture performed in 2i/LIF supplemented ESC medium or FGF4/Heparin supplemented TSC medium, respectively. Therefore, RtL-embryoids were generated as described before and seeded onto gelatin-coated cell culture plates, in either ES or TS medium, where they were allowed to settle and attach. Aggregates were cultured in these conditions, until cellular outgrowth formed, at which point the cells were dissociated by incubation in TrypLE express (Gibco: 12604-013) and passaged onto a new, gelatin-coated culture plate. Once colonies had formed, they were picked and

cultured in individual mono-cultures, before characterization by IF staining for ES and TS marker.

**Immunofluorescence staining of stem cells derived from RtL-embryoids.** For IF staining cells were washed twice with PBS, fixed for 10 min with 4% formalin (Sigma–Aldrich: 100496), washed twice and permeabilized with Triton X-100 ((AppliChem: A4975); 0.5% in PBS). Blocking was performed in blocking solution (2% BSA (Sigma–Aldrich: A9647), 0.1% Triton X-100 (AppliChem: A4975) in PBS) for 1 h at room temperature. Incubation in primary antibody was performed in blocking buffer o/n at 4 °C. Cells were washed three times with PBS before incubation with secondary antibody in blocking buffer for 2 h at room temperature and protected from light. Cells were washed three times with PBS and kept protected from light before imaging. Staining of nuclei was performed for 10 min using Hoechst33342 (Invitrogen: H1399; 1 µg/ml) in PBS. Primary antibodies used and dilutions: Goat-polyclonal anti-CDX2 (Santa Cruz: sc-19478; 1:200), Rabbit-polyclonal anti-EOMES (Abcam: ab23345; 1:400), Mouse-polyclonal anti-OCT4 (Santa Cruz: sc-5279; 1:300), Rabbit-polyclonal anti-NANOG (ReproCell: RCAB002P-F; 1:300). Secondary antibodies used and dilutions: Donkey polyclonal secondary antibody to Goat IgG-H&L Alexa Fluor 594 (Abcam: ab150132; 1:500), Goat-polyclonal secondary antibody to Rabbit IgG-H&L Alexa Fluor 594 (Invitrogen: A-11012; 1:500), Donkey polyclonal secondary antibody to Mouse IgG-H&L (Alexa Fluor 594) (Invitrogen: A-21203; 1:500).

**Imaging and processing.** Confocal images of RtL-embryoids were acquired using VisiScope spinning disk confocal microscope (Visitron) with a ×20, ×40, or ×63 objective. Images in Supplementary Fig. 1b, c as well as Supplementary Fig. 2k - n were acquired using an inverted microscope DM IRB (Leica) with a ×10 objective. Processing and merging of images was performed using the open-source image processing package Fiji (version 1.53c).

**Cellular reprogramming of 5 Factor ESCs in 2D mono-culture and IF staining against CD40.** 5-Factor ESCs were seeded onto gelatin-coated cell culture dishes in ES-medium and cultured at 37 °C and 7.5% CO$_2$ until small colonies had formed. Transgene induction and reprogramming was initiated by changing the cell culture medium to TS medium, consisting of 70% feeder-MEF conditioned TS medium and 30% freshly prepared TS medium, supplemented with FGF4 (25 ng/ml), Heparin (1 µg/ml) and DOX (2 µg/ml). Cells were cultured under these conditions for 3 days, before omitting DOX from the culture medium, thereby stopping transgene expression. The culture medium was replaced with fresh culture medium every 48 h. For FACS and subsequent scRNA-Seq analysis cells were cultured for another 24 h without DOX supplementation, before harvesting by washing with PBS and incubation in StemPro Accutase Cell Dissociation Reagent (Gibco: A1110501). Once cells had dissociated, the reaction was stopped by adding TS culture medium to the cells. The single-cell suspension was pellet by centrifugation and washed twice with PBS before staining for CD40. Staining for CD40 (Goat-polyclonal anti-CD40; R&D Systems: AF440) was performed in FACS buffer (2% FCS in PBS; Dilution: 1:300) for 30 min on ice. Cells were washed three times with FACS buffer, resuspended in FACS buffer with secondary antibody (Chicken anti-Goat IgG (H + L) Cross-Absorbed Secondary Antibody, Alexa Fluor 647; Invitrogen: A-21469; Dilution: 1:300) and incubated for 30 min on ice in the dark. Cells were washed three times with FACS buffer and kept in the dark until sorting.

**Cellular reprogramming of iGATA6 ESCs in 2D mono-culture.** For analysis of iGATA6 ESC reprogramming in 2D mono-cultured, iGATA6_mCherry ESCs were seeded onto gelatin-coated cell culture dishes in ES-medium and cultured at 37 °C and 7.5% CO$_2$ until small colonies had formed. Transgene induction and reprogramming was initiated by changing the cell culture medium to XEN cell culture medium supplemented with 2 µg/ml DOX. Cells were cultured under these conditions for 3 days, before omitting DOX from the culture medium, thereby stopping transgene expression. The culture medium was replaced with fresh culture medium every 48 h. For FACS and subsequent scRNA-Seq analysis cells were cultured for another 24 h without DOX supplementation, before harvesting by washing with PBS and incubation in StemPro Accutase Cell Dissociation Reagent (Gibco: A1110501). Once cells had dissociated, the reaction was stopped by adding XEN cell culture medium to the cells. iXEN cells derived from iGATA6_mCherry were not stained before sorting by FACS, as mCherry expression was used the fluorescent marker for sorting. Cells were pelleted by centrifugation, resuspended in FACS buffer, and kept on ice until sorting.

**FACS sorting of cells from RtL-embryoids and 2D mono-culture reprogramming.** Fluorescence-activated cell sorting (FACS) was used to separate cells of each embryo-like compartment for subsequent transcriptional profiling by scRNA-Seq. Therefore, RtL-embryoids were built from Kermit ESCs, 5-Factor ESCs, and a mCherry-transduced iGATA6 ESC line in ratios previously described. A total n of >600 correctly assembled RtL-embryoids were handpicked after reprogramming and depletion from DOX, pooled, and dissociated into a single-cell suspension by incubation for 15 min in StemPro Accutase Cell Dissociation Reagent. After passing through a 40 µm cell strainer (Becton Dickinson), cells were stained against CD40 (Goat-polyclonal anti-CD40; R&D Systems: AF440), a surface protein

expressed on cells of ExE-identity, allowing for fluorescence-activated cell sorting of either GFP, mCherry or Alexa-647 for isolation of Kermit ESCs, iXEN cells or iTSCs, respectively. Staining for CD40 was performed in FACS buffer (2% FCS in PBS; Dilution: 1:300) for 30 min on ice. Cells were washed three times with FACS buffer, resuspended in FACS buffer with secondary antibody (Chicken anti-Goat IgG (H + L) Cross-Absorbed Secondary Antibody, Alexa Fluor 647; Invitrogen: A-21469; Dilution: 1:300) and incubated for 30 min on ice in the dark. Cells were washed three times with FACS buffer and kept in the dark until sorting. Live/dead staining was performed using the Fixable Near-IR Dead Cell Stain Kit (Invitrogen; L34975), according to manufacturer's protocol, by incubation for 15 min at RT. Flow cytometry and sorting were performed using FACS DIVA (version 8.0.1) on BD Aria III (BD Bioscience Pharmingen). For cells obtained from reprogramming in 2D mono-culture the identical FACS setup was used. Kermit ESCs were identified and sorted by GFP signal, iTSCs reprogrammed from 5 Factor ESCs were sorted by Alexa-647 and iXEN cells reprogrammed from iGATA6-mCherry ESCs were identified by mCherry signal.

**Library preparation and sequencing using Smart-Seq2.** Our new index-sorted single-cell transcriptome dataset was based on the Smart-Seq2 protocol[18]. Cells were FACS sorted into eight 384-well plates containing 2.3 µl lysis buffer (Guanidine Hydrochloride (50 mM; Sigma–Aldrich: G3272), dNTPs (17.4 mM; NEB: N0447), SMART dT30VN primer (2.2 µM; IDT) retaining protein expression information for every well to subsequently match with the respective single-cell transcriptomic data in an index sorting approach. Plates were sealed and stored at −80 °C until further processing. Smart-Seq2 libraries were finally generated on a Tecan Freedom EVO and Nanodrop II (BioNex) system as previously described[18]. In short, lysed cells were incubated at 95 °C for 3 min. 2.7 µl RT mix containing SuperScript II buffer (Invitrogen: 18064071), 9.3 mM DTT, 370 mM Betaine (Sigma–Aldrich: B0300), 15 mM MgCl2 (Sigma–Aldrich: 63069), 9.3 U SuperScript II RT (Invitrogen: 18064071), 1.85 U recombinant RNase Inhibitor (Takara: 2313 A), 1.85 µM template-switching oligo (Eurogentec) was aliquoted to each lysed cell using a Nanodrop II liquid handling system (BioNex) and incubating at 42 °C for 90 min and 70 °C for 15 min. 7.5 µl preamplification mix containing KAPA HiFi HotStart ReadyMix (KAPA: 7958935001) and 2 µM ISPCR primers (IDT) was added to each well and full-length cDNA was amplified for 16 cycles. cDNA was purified with 1× Agencourt AMPure XP beads (Beckman Coulter: A63882) and eluted in 14 µl nuclease-free water (Invitrogen: 15667708). Concentration and cDNA size was checked for select representative wells using a High Sensitivity DNA5000 assay for the Tapestation 4200 (Agilent: 5067-5592). cDNA was diluted to an average of 200 pg/µl and 100 pg cDNA from each cell was tagmented by adding 1 µl TD and 0.5 µl ATM from a Nextera XT DNA Library Preparation Kit (Illumina: FC-131-1096) to 0.5 µl diluted cDNA in each well of a fresh 384-well plate. The tagmentation reaction was incubated at 55 °C for 8 min before removing the Tn5 from the DNA by adding 0.5 µl NT buffer per well. 1 µl well-specific indexing primer mix from Nextera XT Index Kit v2 Sets A-D and 1.5 µl NPM was added to each well and the tagmented cDNA was amplified for 14 cycles according to manufacturer's specifications. PCR products from all wells were pooled and purified with 1× Agencourt AMPure XP beads (Beckman Coulter) according to the manufacturer's protocol. The fragment size distribution was determined using a High Sensitivity DNA5000 assay for the Tapestation 4200 (Agilent) and library concentration was determined using a Qubit dsDNA HS assay (Thermo Fischer). Libraries were clustered at 1.4 pM concentration using High Output v2 chemistry and sequenced on a NextSeq500 system SR 75 bp with 2*8 bp index reads. Single-cell data were demultiplexed using bcl2fastq2 v2.20.

**Single-cell RNA-Seq raw data processing.** Following sequencing by the Smart-Seq2 method[18], RNA-Seq libraries were subjected to initial quality control using FASTQC (version 0.11.7) implemented in a scRNA pre-processing pipeline (docker image and scripts available at [https://hub.docker.com/r/pwlb/rna-seq-pipeline-base/] (version 0.1.1); [https://bitbucket.org/limes_bonn/bulk-rna-kallisto-qc/src/master/] (version 0.2.1)). Next, raw reads were pseudoaligned to the mouse transcriptome (GRCm38, Gencode vM16 primary assembly) using Kallisto with default settings (version 0.44.0)[84]. Based on the pseudo alignment estimated by Kallisto, transcript levels were quantified as transcripts per million reads (TPM). TPM counts were filtered for mitochondrial- pseudo- and ribosomal genes before imported into R using tximport (version 1.16.1)[85] and transcript information was summarized on gene-level. We imported the resulting dataset of 29,960 features across 1931 samples and performed the downstream analysis using the R package Seurat (version 3.1.2)[86].

**Data quality control.** We excluded the cells with less than 2500 expressed genes and less than 500,000 aligned sequencing reads. After filtering the 3D co-culture dataset contains 29,960 features across 961 samples. The 3D co-culture and 2D culture combined dataset contains the same feature number across 1385 samples.

**Dataset integration and dimensionality reduction of scRNA-seq data.** Log-Normalization (Seurat function) was applied before downstream analysis. The original gene counts for each cell were normalized by total transcript counts, multiplied by 10,000 (TP10K), and then log-transformed by log10(TP10k + 1).

Next, the genes with the highest cell-to-cell variability in the dataset were determined by calculating the top 2000 most variable genes by selecting the 'vst' method of the 'FindVariableFeatures' function in Seurat (version 3.1.2). After scaling, the dimensionality of the 3D co-culture data was reduced to 10 principal components (PCs) and 3D co-culture and 2D mono-culture combined dataset was reduced to 4 PCs that were used as input for UMAP representation.

**Clustering and cluster annotation**. The SNN-graph-based Louvain clustering of the 3D co-culture dataset was performed using a resolution of 0.2. Clusters were annotated by comparing cluster marker genes with public sources. 2D culture clusters were annotated via culture origin.

**Differential expression tests and cluster marker genes**. DE tests were performed using FindMarkers/FindAllMarkers functions in Seurat (version 3.1.2) with default two-sided nonparametric Wilcoxon Rank Sum test with Bonferroni correction using all genes in the dataset. Cluster marker genes with >1.5 log-fold changes were regarded as significantly differentially expressed DEGs. Cluster marker genes were identified by applying the DE tests for upregulated genes between cells in one cluster to all other clusters in the dataset. Top-ranked genes (p-values) from each cluster of interest were extracted for further illustration. All other function arguments were set to default. For the differential expression tests between 3D co-culture and 2D mono-culture equivalents default settings were used.

**Subset analysis of the VE-like, Epi-like, and ExE-like cells**. The VE-like, Epi-like, and ExE-like cells space was examined by subsetting the 3D co-culture dataset. The Variable gene selection was repeated (top 2000 variable genes), scaling of the VE-like, Epi-like, and ExE-like cells were performed with the ScaleData function of Seurat (version 3.1.2). The dimensionality of the VE-like-cell data was then reduced to 7 PCs, Epi-like cells data to 4 PCs, and ExE-like-cell data to 10 PCs, these reductions served as input for the UMAP calculations. The SNN-graph-based Louvain clustering of the cells was performed using a resolution of 0.2 for the VE-like-cell space, 0.3 for the Epi-like-cell space and 0.1 for the ExE-like-cell space.

**Signature enrichment analysis**. A gene signature enrichment analysis using the 'AUCell' method[87] was implemented in the package (version 1.10.1) in R. We applied the package to link observed Stem cell clusters to existing studies. The resulting AUC values were normalized the maximum possible AUC to 1 and subsequently visualized in violinplots.

**Integration of the reference datasets**. The collection of reference datasets was kindly supplied as a Seurat object by Posfai et al.[24]. The received object was updated for analysis in R (version 4.0.3) with Seurat (version 4.0.1). The object was subsetted for cells of datasets generated by Chen et al.[26], Mohammed et al.[27], Posfai et al.[25], and Pijuan-Sala et al.[28]. The integrated Seurat object was split into the respective datasets. Normalization and variable feature selection were performed independently as described above. Repeatedly variable features were identified across the datasets using the function 'SelectIntegrationFeatures'. Scaling and PCA were performed based on these features for each of the datasets. Integration anchors were identified using 'FindIntegrationAnchors' using 'rpca' reduction and by setting 'k.anchor' = 20. The integration was performed with the 'IntegrateData' function by setting 'k.weight' = 45. The integrated data were scaled, and dimensionality reduction by PCA and UMAP was performed.

**Label transfer between the integrated reference dataset and the RtL-embryoids dataset**. The RtL-embryoid dataset was updated for analysis in Seurat (version 4.0.1.) The 'FindTransferAnchors' function was used to determine anchors between our dataset and the integrated reference datasets. The 'MapQuery' function was used to map our dataset onto the UMAP of the reference dataset. Since non-perfectly overlapping cell identities were expected between the RtL-embryoid dataset and the reference datasets, a 'de-novo visualization' by merging the PCA embeddings of the two datasets and calculation of a new UMAP was performed.

**Confusion matrix**. For visualization of the results of the Label Transfer described above, the percentage of cells from the RtL-embryoid dataset mapped to the respective cells in the reference dataset was calculated and visualized using pheatmap (version 1.0.12)

**GO enrichment analysis**. Significantly DEGs between each cluster were identified by the FindAllMarkers function from the Seurat package using the Wilcoxon Rank Sum test. The top 100 DEG sorted by adjusted p-value were used for the GO enrichment test by R package/ClusterProfiler (version 3.14.3)[88].

**Transcription factor enrichment analysis**. Significantly DEGs between each cluster were identified by the FindAllMarkers function from the Seurat package using Wilcoxon Rank Sum test. All upregulated DEGs were analyzed by Enrichr ChIP enrichment analysis (ChEA16) [http://amp.pharm.mssm.edu/Enrichr/] for

determining the transcription factors that could control the expression of these genes[89,90].

**NicheNet analysis**. The NicheNet approach predicts which ligands formed by one cell type regulate the expression of which target genes in another cell type. Ligand-target links are inferred by RNA expression data of interacting cells with current knowledge on signaling and gene regulatory networks.

NicheNet can address the question of which ligands produced by a sender cell are the most active in impacting gene expression in the receiver cell (i.e., ligand activity analysis)[68]. To achieve this, NicheNet assesses how well these ligands predict the examined changes in gene expression and ranks them according to this. After ligand ranking, NicheNet infers active ligand-target connections by looking for genes that are impacted in the receiving cell type and have a high possibility to be regulated by the prioritized ligands[68].

**NicheNet Intercellular communication analysis**. For the NicheNet analysis (version 1.0.0)[68], we performed a NicheNet ligand activity analysis for every cell type separately (VE-like, Epi-like, ExE-like cells). We accepted all genes as potential ligands that were expressed in >10% of the individual sender cell type and which matched at least one receptor from the genes that were considered as differentially expressed in the receiver cells. As background, we considered all other genes that are not differentially expressed. For ligand prioritization, we selected the top 6 genes with the highest Pearson correlation coefficient for all three cell types. To make the activity scores in all three settings comparable overall ligands, z-score normalization of the Pearson correlation coefficients were performed after combining all ligands for the different cell types together. For those ligands, the corresponding receptors are indicated in the ligand-receptor heatmap. The indicated score is assigned to the weight of the interaction between the ligand and receptor in the integrated weighted ligand signaling network. In the ligand-target heatmap, we show regulatory potential scores for interactions between the 6 top-ranked ligands and 100 top target genes for each cell type.

**Data visualization**. In general, the R packages Seurat and the ggplot2 package (version 3.3.2)[91] were used to generate figures. For visualization of bar plots and quantitative data, we used GraphPad Prism (version 8.0.2). Schematics were generated using Powerpoint (version 2110). Flow cytometry data were visualized by using FlowJo (version 10.6.1).

**Reporting summary**. Further information on research design is available in the Nature Research Reporting Summary linked to this article.

## Data availability

Raw sequencing data of mouse Smart-seq2 generated in this study have been deposited in the Gene Expression Omnibus (GEO) database under accession code GSE188394. Additionally, data are deposited via FASTGenomics [https://beta.fastgenomics.org/d/200474]. The FASTGenomics platform also provides Seurat objects of the datasets generated in this study. The publicly available datasets analyzed during the current study are available from the GEO- and ArrayExpress repository. GSE84892[25] GSE74155[26] GSE100597[27] E-MTAB-6967[28] The mouse genome used for kallisto alignment is available from the GENCODE Project. GRCm38, Gencode vM16 primary assembly. Source data are provided with this paper.

## Code availability

All the computational analyses were performed using R programming languages. Scripts of key steps can be found at [http://github.com/schultzelab/Rosette-to-Lumen-stage-embryoids]. Additionally, our code to reproduce the analysis can be accessed via FASTGenomics [https://beta.fastgenomics.org/analyses/detail-analysis-9301d957888c40538020f90428bcc763#Result&analysis].

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

## Acknowledgements

We kindly thank Gaby Beine, Angela Egert, Andrea Jäger, Heidi Theis, and Michael Kraut for perfect technical assistance. Further, we would like to thank the Microscopy Core Facility of the Medical Faculty at the University of Bonn for providing support und instrumentation funded by the Deutsche Forschungsgemeinschaft (DFG, German Research Foundation)—Projektnummer 388169927. This work was funded by grants of the Deutsche Forschungsgemeinschaft (DFG) SCHO 503/20-1 to HS and SCHU 950/8-1 to J.L.S. J.L.S. was further supported by ImmunoSensation2 EXC2151/1, Healthy Diet for a Healthy Life (JPI-HDHL; project 529051018), BMBF-funded excellence project Diet-Body-Brain (DietBB) under grand number 01EA1809A, GRK 2168 of the DFG grant number 272482170-, and by the Helmholtz Alliance 'Aging and Metabolic Programming, AMPro'.

## Author contributions

Conceptualization: J.L.S. and H.S.; Methodology: J.L., A.H., F.K., A.C.A., T.P., L.B., L.H., T.H., A.K., T.E., J.L.S., and H.S.; Software: A.H., K.B., L.H., N.R., and J.L.S.; Formal analysis: A.H., J.L., L.H., A.S., K.H., L.B., and M.B.; Investigation: J.L., A.H., L.H., L.B., Y.R., L.H.Y., J.L.S., and H.S.; Resources: C.K., Data Curation: A.H., K.H., M.B., T.U., and J.L.S.; Writing—original draft J.L., A.H., J.L.S., and H.S.; Visualization: A.H., J.L., and L.H.; Supervision: J.L.S. and H.S.; Funding: Acquisition J.L.S. and H.S.

## Funding

## Competing interests

The authors declare no competing interests.
