## [Peer Review File · Nature Communications]

Induction of Rosette-to-Lumen stage embryoids using reprogramming paradigms in ESCsReviewers' Comments:

Reviewer #1:

Remarks to the Author:

In this manuscript, Langkabel et al. show that mixing three genetically engineered lines of ESCs equipped for the induction of specific transcription factors is sufficient for the three lines to organize into structures reflecting the inner part of the ~E5 embryo. This model might represent an important tool to understand, for example, the role of the extra embryonic tissues in the exit of pluripotency.

It would be important to precisely depict and name the structures that are formed and the ones that do not form as this is crucial in order to understand what these structures can model. In order to clarify the tissues that are modelled and the ones that are not, the authors should make a schematic of the early pre- and post-implantation embryo including the names of the main tissues and highlighting the ones that form while shading the tissues that are not.

In order to clarify the specific importance of their model in the overall landscape of embryo models, the authors should give a name/acronym to their model that would reflect the tissue analogs that are formed and the stage. The authors should avoid using the term synthetic embryo-like structure that does not explain what is being modelled as compared to previously developed embryo models, and that is ambiguous as these structures are not synthetic but rather formed using stem cells that recapitulate aspects of development.

This model does not form the outer tissues of the embryo, namely the EPC, the trophoctoderm, the parietal endoderm, and, as such, cannot model processes of implantation in utero. The claims relative to figure 7 are thus incorrect and misleading. The implantation of the embryo and the establishment of the embryo-maternal interface occur via tissues that do not form in this model.

Textual changes:

"Can organise into early embryo-like structures". This sentence is not precise enough and should be replaced by for example "can organise into structures reflecting aspects of the inner regions of the early post-implantation embryo."

The term symmetry breaking is not correct and should be removed. Symmetry breaking occurs around the morula stage, not at the early post-implantation stage. What is observed is patterning. Symmetry breaking requires a homogeneous population and surrounding environment as a starting point and is indeed often misused.

"Resembling natural murine embryos." This sentence is not precise enough and should be replaced by for example "can organise into structures reflecting aspects of the inner regions of the post-implantation embryo."

The DKK1 hypothesis is interesting and has already been proposed in the following articles PMID: 25910836, PMID: 24859004, and PMID: 32367046. This should be clearly mentioned and referenced.

Major additional experiments:

The generation and analysis of the scRNAseq dataset is important and should be completed by merging it with previously published datasets of the mouse embryo as done in Posfai et al. PMID: 33420491 figure 3. This strictly in silico analysis would allow to better evaluate the quality of the reprogramming, the stage of the different compartments, and their level of synchronization in time.

The evaluation of the patterning of the anterior-posterior patterning (figure 3) should be quantified

over a larger number of structures. The results should be reported by measuring (1) the number of cells positive for AVE markers including but not restricted to Lefty and (2) the percentage of structures that undergo a certain level of patterning according to specific quantitative criteria to be determined by the authors.

The power of the model might lie in understanding the role of extra-embryonic tissues in the exit of pluripotent around the rosette stage. As such, it is crucial to validate the switch of transcription factor expression that occurs, as extensively described in the literature including by immunofluorescence of Nanog, Oct4, Klf4, Esrrb, Oct6, Otx2. Immunofluorescence for additional molecules including Podxl, pERK should also be included. Note that the transition of the pluripotent compartment has been extensively characterised in PMID: 32367046, which would provide a roadmap for benchmarking the model.

The putative spatial segregation of the cluster 1 and 2 of trophoblasts should be validated using immunofluorescence.

The claims relative to figure 7 are incorrect and misleading. The implantation of the embryo and the establishment of the embryo-maternal interface occur via tissues that do not form in this model. As such this model does not have the capacity to model aspects of implantation. This data should be removed.

Nicolas Rivron

Reviewer #2:

Remarks to the Author:

In this manuscript by Langkabel and colleagues describes an innovation in the construction of synthetic mouse embryos. Whereas other researchers have combined three different types of embryo-derived stem cell line (ES, TS, and XEN), the authors here use only mouse ES cells, which could streamline the process for some labs. Since ES cells typically produce only the fetal lineage, and not the extraembryonic cell types normally provided by TS and XEN cells, the authors use two engineered ES cell lines, capable of overexpressing key trophoblast and extraembryonic endoderm transcription factors. They show that combining the three ES cell lines, and then inducing expression of the extraembryonic lineage factors leads to formation of structures bearing similarities to mouse peri-implantation stage embryos.

Important questions:

I am finding myself to be a little underwhelmed by the observation that ES cell lines induced to adopt alternative cell fates bear the expected transcriptional signatures of those cell types when co-cultured and then deconstructed. I guess it's a good confirmation, but I found myself wanting to know if co-culture enabled these cell lines to become even more like their physiological in vivo counterparts than when the lines are induced and cultured separately in 2D format. In other words, do the authors have the negative control data (transcriptional analysis of each cell line, induced separately as monolayers) and/or positive control data (embryos)? Was there any difference in the transcriptional signatures in correctly organized and non correctly organized structures?

Related query: can ES, XEN, and TS cells be derived from these structures? Is there functional evidence that these cocultured cell lines adopt the expected cell fates (and not just transcriptional evidence)?

Could the authors provide evidence that CD40 is lineage-specific in this context?

Could the authors comment on how the proportion of synthetic embryo-like structures with asymmetric AVE-like zone compares with asymmetry of such structures created by other protocols (in other publications)?

Related issue: the evidence of embryo patterning is quite superficial. Could the authors use standard assays for assessing embryo patterning (not single cell analysis in deconstructed structures, but in situ with patterning markers used by embryologists). Does T come on opposite the LEFTY1? Is there a gradient of phospho-MAPK in the EXE-like zone?

Could the authors please move the graphs shown in Supp Fig 2A and B to the main figures. This contains an important accounting of the outcome efficiencies critical to supporting the authors' claims

Could the authors provide some kind of control for the uterine implantation analysis? As I recall, even beads of oil can cause decidualization, raising the possibility that these embryo structures aren't engaging a particularly unique outcome. What happens when periimplantation embryos are transferred to recipient females in the same manner?

Minor points:

Paragraph 1 of Intro: "conceptus as a whole is surrounded by the parietal yolk sac, Reichert's membrane, and a layer of trophoblast giant cells."

In fact, the extraembryonic tissues are part of the conceptus; they do not surround the conceptus. Moreover, Reichert's membrane is comprised of the parietal endoderm and the trophoblast giant cells.

Paragraph 1 of Results, "In recent years, we and others demonstrated that ESC can be reprogrammed to bona fide TSC and XEN cells (Wamaitha et al., 2015; Kaiser et al., 2020)." Should include additional relevant references (e.g., Niwa et al., 2005).

Did the authors mix up distal and proximal EXE definitions? I know many papers put it this way, but it seems like the EXE's axis should be inverted with respect to the epiblast's axis, such that proximal epiblast and proximal EXE are neighbors.

Figure 1 B-F could be reorganized. It was quite confusing that some panels are vertical and some are horizontal. One possibility is to eliminate some of the redundancy of the various color combinations.

Figure 1 would be improved by including the quantification here – how many embryo-like structures resulted in correctly localized lineage-specific markers?

Figure 3F – the color combos are hard to see – could the authors please separate the channels and show at least LEFTY1 alone?

Figure 5D needs a scale bar of expression level (violin plot better)?

Figure 5E – which is proximal and which is distal?

Figure 6A – I wasn't clear what was meant by "ligand activity"

Why would implantations be expected to resemble E6.5 embryos after a week of in vivo development? What is observed at an earlier time point?

Fig. 7D – was Cherry detectable?

Reviewer #3:

Remarks to the Author:

The manuscript by Langkabel, Horne et al. describes a strategy for genetic modification of ESCs leading to induction of either TSC-fate or XEN-fate, and generation of embryo-like structures using mixtures of modified cells in a 3D co-culture. This approach would offer important advantages over existing, more complex methods for generating synthetic embryo-like structures.

The authors largely rely on scRNA-seq to explore and validate the identities of cells of different compartments in the embryo-like structures. These analyses are based on a rather limited number of cells and biological replicates. However, in my opinion this is sufficient given the relative simplicity of the experimental model. The results are for the most part very clear and are in line with the conclusions made by the authors. The limitations of the study are also adequately described. Finally, the paper is very well written, and the quality of images is good. Altogether, I would recommend the paper for publication with the following minor remarks:

Figure 3D - differences in expression for some of the markers are not very clear. which of the differences are statistically significant?

Figure 4C are these markers expressed in the other clusters besides Epi?

Figure S3: The term "Top variable genes" is somewhat misleading, as it can also refer to the noisiness of expression. "30 most differentially expressed (DE)" would be more appropriate and this is also used in the main text.

Methods- Data quality control

The authors describe the following approach "We excluded the cells with less than 2500 expressed genes and less than 500,000 transcripts." - What exactly is counted as a transcript? If this refers to aligned sequencing reads, the term should be changed accordingly.

AUTHORS RESPONSES TO EVIEWER COMMENTS

Dear Reviewers, dear Nicolas,

Thanks for your comments and suggestions. Please find enclosed a point-by-point response to all concerns and issues raised. We performed additional experiments, re-arranged the manuscript and included newly generated and computed datasets. Your input helped us to improve the story substantially.

Please see the individual points addressed below, the main text ist highlighted where we added, amended and changed passages.

Best regards

H. Schorle

Reviewer #1 (Remarks to the Author):

In this manuscript, Langkabel et al. show that mixing three genetically engineered lines of ESCs equipped for the induction of specific transcription factors is sufficient for the three lines to organize into structures reflecting the inner part of the ~E5 embryo. This model might represent an important tool to understand, for example, the role of the extra embryonic tissues in the exit of pluripotency.

It would be important to precisely depict and name the structures that are formed and the ones that do not form as this is crucial in order to understand what these structures can model. In order to clarify the tissues that are modelled and the ones that are not, the authors should make a schematic of the early pre- and post-implantation embryo including the names of the main tissues and highlighting the ones that form while shading the tissues that are not.

Author's reply:

We agree with the reviewer, that it is important to precisely depict and name the structures represented by this model and the ones that are not. Therefore, we added a schematic, highlighting the tissues formed and shading absent tissues (Fig 1F). We furthermore provided additional evidence supporting rosette and lumen formation in Epi- and ExE-like compartments (Fig. 1G and H). An example for the fusion of the lumen has been added as well, however, such structures were rarely observed and could not be studied in detail (Fig. 1I). Nevertheless, this observation can be

regarded additional evidence for the developmental potential of RtL-embryoids.

In order to clarify the specific importance of their model in the overall landscape of embryo models, the authors should give a name/acronym to their model that would reflect the tissue analogs that are formed and the stage. The authors should avoid using the term synthetic embryo-like structure that does not explain what is being modelled as compared to previously developed embryo models, and that is ambiguous as these structures are not synthetic but rather formed using stem cells that recapitulate aspects of development.

Author's reply:

We agree with the reviewer, that the term “synthetic embryo-like structure” does not properly reflect the exact development stage represented by the model presented in this study. As the additional experiments proposed by the reviewer were indeed very useful to determine the exact developmental stages reflected by the embryo-model, we decided to name the structures RtL-embryoids (Rosette-to-Lumen-embryoids). Furthermore, we decided to add an additional paragraph to the discussion, in which we placed the model presented here in the context of other related protocols, as proposed by the Editor. Regarding the term “synthetic”, we agree with the reviewer’s opinion that this term is ambiguous, however, as Rossant & Tam recently discussed (PMID: 33667412), such “synthetic models” would require that “...new genetic switches or synthetic pathways have been programmed into the starting cells...” As such pre-programmed genetic switches can be regarded a pre-requisite for the model presented in this study, we think the term “synthetic” can be used in this instance. We therefore added an additional paragraph in the discussion putting the term “synthetic” into context. The paragraph can now be found in Line 534 – 548.

This model does not form the outer tissues of the embryo, namely the EPC, the trophoctoderm, the parietal endoderm, and, as such, cannot model processes of implantation in utero. The claims relative to figure 7 are thus incorrect and misleading. The implantation of the embryo and the establishment of the embryo-maternal interface occur via tissues that do not form in this model.

Author's reply:

The reviewer correctly noted that the tissues required for implantation and the establishment of the embryo-maternal interface are not formed in the model presented in this study. We agree with the reviewer’s comment, that the claim of implantation of RtL-embryoids cannot be assessed and compared to the natural process of implantation.

Textual changes:

“Can organise into early embryo-like structures”. This sentence is not precise enough and should be replaced by for example “can organise into structures reflecting aspects of the inner regions of the early post-implantation embryo.”

Author’s reply:

We agree and changed the respective sentence in the abstract to: “ESCs in these cultures self-organize into elongated, compartmentalized embryo-like structures reflecting aspects of the inner regions of the early post-implantation embryo”

Line 28 – 29

The term symmetry breaking is not correct and should be removed. Symmetry breaking occurs around the morula stage, not at the early post-implantation stage. What is observed is patterning. Symmetry breaking requires a homogeneous population and surrounding environment as a starting point and is indeed often misused.

Author’s reply:

We agree and removed the term “symmetry breaking” from the sentence in the Introduction, which now reads:

“These structures exhibit key hallmarks of embryogenesis, such as patterning events and they were shown to implant *in uteri* upon transplantation².”

Line 54 - 55

“Resembling natural murine embryos.” This sentence is not precise enough and should be replaced by for example “can organise into structures reflecting aspects of the inner regions of the post-implantation embryo.”

Author’s reply:

We changed the respective sentence in the Introduction to:

“Using an easy to generate agarose-based 3D mold, we demonstrate that transgene-induced cellular reprogramming and self-organization into embryo-like tissues occur in parallel. The resulting structures resemble the inner regions of early post-implantation murine embryos at E4.5 – E5.5”

Line 67 - 70

The DKK1 hypothesis is interesting and has already been proposed in the following articles PMID: 25910836, PMID: 24859004, and PMID: 32367046. This should be clearly mentioned and referenced.

Author's Reply:

We addressed the requirements for textual changes as proposed by the reviewer and changed the respective passages accordingly. References for PMID: 25910836, PMID: 24859004, and PMID: 32367046 have been added and mentioned in the text, which now reads:

„ Furthermore, DKK1 expression has been demonstrated to be initiated in the mature primitive endoderm starting around E4.5^{39,68} and was proposed to indicate the completion of PrE maturation, repressing WNT signalling and allowing for OTX2 induction, rosette formation and pluripotency progression³⁹.”

Line 447 – 450

Major additional experiments:

The generation and analysis of the scRNAseq dataset is important and should be completed by merging it with previously published datasets of the mouse embryo as done in Posfai et al. PMID: 33420491 figure 3. This strictly in silico analysis would allow to better evaluate the quality of the reprogramming, the stage of the different compartments, and their level of synchronization in time.

Author's Reply:

We agree with the reviewer's comment, that the scRNA-Seq dataset should be merged and completed with previously published datasets of the mouse embryo. Therefore, we aligned our dataset with several scRNA-Seq datasets of murine embryogenesis, obtained from different studies, as done in Posfai et al.. Cells of RtL-embryoids clustered close with cells of their embryonic equivalents and the comparison supported the presumed developmental stage around E4.5 – E5.5. Additional Figures can now be found in Figure 2F, Supplementary Fig. 2H and 2I. Textual changes describing the comparison of scRNAseq datasets can now be found in line 185 – 198.

The evaluation of the patterning of the anterior-posterior patterning (figure 3) should be quantified over a larger number of structures. The results should be reported by measuring (1) the number of cells positive for AVE markers including but not restricted to Lefty and (2) the percentage of structures that undergo a

certain level of patterning according to specific quantitative criteria to be determined by the authors.

Authors' Reply:

We performed IF stainings against additional DVE/AVE markers and were able to detect a subpopulation of VE-like cells expressing the DVE/AVE marker EOMES. EOMES functions in the visceral endoderm as an instructive regulator of the transcriptional program controlling anterior-posterior axis formation (Nowotschin et al., 2013; PMID: 23651855). Furthermore, we were able to detect expression of LEFTY1 and OTX2 in these EOMES+ VE-like cells, strengthening our hypothesis, that the VE-like compartment of the embryo-model presented in this study does induce a DVE/AVE-like cell population. A schematic describing formation of DVE/AVE and the additional IF stainings for EOMES, OTX2 and LEFTY1 can now be found in Figure 3 C, F, G, H.

To determine the exact number of DVE/AVE-like cells per aggregate, we performed Z-Stack imaging in combination with LEFTY1/EOMES/Phalloidin stainings, which would allow for identification of single cells as well as double positivity for both AVE marker. However, this experimental approach was hindered due to technical hardware and software difficulties and could not be realized. We do however think that the detection of an EOMES+/OTX2+/LEFTY1+ population that is restricted to the EmVE-like tissue provides sufficient proof to support our claim of the induction of a DVE/AVE-like cell population. Furthermore, we provide additional quantification regarding the localization of such a LEFTY1+ DVE/AVE-like population in either distal-, transition- or anterior- position. In the majority of RtL-embryoids such a DVE/AVE-like population was detected in a distal or transition position. Only few aggregates displayed such a population in an anterior position. Examples for assessed LEFTY1+ positions can now be found in Figure 3I and respective quantification is provided in Figure 3J. Additionally, we added an example of RtL-embryoids showing weak contribution to the ExE-like compartment, resulting in a failure to restrict Lefty1 expression to the distal part of the EmVE, most likely due to insufficient BMP mediated inhibition from the ExE-like cells (Supplementary Figure 3B).

As we did not observe a complete anterior-posterior patterning within the epiblast in response to this DVE/AVE-like population and did not detect T in a presumptive posterior epiblast, we reduced our claim of an “AVE like patterning” and discussed the formation of a DVE/AVE like population in the EmVE.

The respective textual changes to the manuscript can now be found in the section describing the VE-like compartment from line 257 – 272.

The power of the model might lie in understanding the role of extra-embryonic tissues in the exit of pluripotent around the rosette stage. As such, it is crucial to validate the switch of transcription factor expression that occurs, as extensively described in the literature including by immunofluorescence of Nanog, Oct4, Klf4, Esrrb, Oct6, Otx2. Immunofluorescence for additional molecules including Podxl, pERK should also be included. Note that the transition of the pluripotent compartment has been extensively characterised in PMID: 32367046, which would provide a roadmap for benchmarking the model.

Author's reply:

The remark of the reviewer proved to be extraordinarily helpful in characterizing the embryo-model presented in this study. We performed IF staining to detect presence/absence of NANOG, OCT4, KLF4, ESRRB, OCT6, OTX2, pERK, PODXL and general cellular architecture using Phalloidin staining against actin. We were able to show that the aggregates undergo rosette and lumen formation in both, EPI- and ExE-like compartments, as indicated by Phalloidin or PODXL stainings. At rosette stage EPI-like cells were found to express KLF4, ESRRB and OTX2. At lumen stage, KLF4 and ESRRB were found to be downregulated / absent in most Epi-like cells and we were able to detect weak expression of OCT6 in some OTX2+ cells. In this manner, the Epi-like compartment of RtL-embryoids did progress from a naïve- to primed- pluripotency state. Additionally, we were able to detect single pERK+ cells within the Epi-like compartment, again, showing similarities to the natural murine epiblast. We did however observe steady expression of NANOG throughout the culture period, indicating that RtL-embryoids fail to downregulate NANOG and do not fully recapitulate the roadmap of pluripotency progression as described for murine embryogenesis. The respective IF stainings, UMAPs and Heatmaps showing expression of the marker genes can now be found in Figure 4 A, D, E, F, G, H, I and J. Corresponding textual changes have been made in the paragraph describing the Epi-like compartment (Line 288 – 319) and PMID: 32367046 has been added as a reference.

The putative spatial segregation of the cluster 1 and 2 of trophoblasts should be validated using immunofluorescence.

Author's reply:

We were able to demonstrate the putative spatial segregation of trophoblast cluster 1 and 2 by IF staining against pERK, as proposed by Reviewer 2, revealing a pattern of MAPK activity within the ExE-like tissue similar to observations made during mouse embryogenesis between E5.5 and E6.5⁶¹. Additionally, comparison between 2D mono- and 3D co-culture-based reprogramming revealed high transcriptional similarity between ExE-like subcluster 1 and iTSCs generated from 5 Factor ESCs in TS medium supplemented with FGF4/Heparin, highlighting the stem cell characteristics of ExE-like subcluster 1. The respective IF staining and a UMAP displaying the 2Dvs3D comparison can now be found in Figure 5 F, G and the passage describing the ExE-like compartment was changed accordingly and can now be found in line 390 - 414.

The claims relative to figure 7 are incorrect and misleading. The implantation of the embryo and the establishment of the embryo-maternal interface occur via tissues that do not form in this model. As such this model does not have the capacity to model aspects of implantation. This data should be removed.

Nicolas Rivron

Author's reply:

As mentioned before, the reviewer's comment regarding missing tissues required for implantation is correct. We agree with the reviewer's opinion, that reliable modelling of implantation would require formation of all respective tissues needed. Therefore, we removed the data from the manuscript.

--

Reviewer #2 (Remarks to the Author):

In this manuscript by Langkabel and colleagues describes an innovation in the construction of synthetic mouse embryos. Whereas other researchers have combined three different types of embryo-derived stem cell line (ES, TS, and XEN), the authors here use only mouse ES cells, which could streamline the process for some labs. Since ES cells typically produce only the fetal lineage, and not the extraembryonic cell types normally provided by TS and XEN cells, the authors use two engineered ES cell lines, capable of overexpressing key trophoblast and extraembryonic endoderm transcription factors. They show that combining the three ES cell lines, and then inducing expression of the extraembryonic lineage factors leads to formation of structures bearing similarities to mouse peri-implantation stage embryos.

Important questions:

I am finding myself to be a little underwhelmed by the observation that ES cell lines induced to adopt alternative cell fates bear the expected transcriptional signatures of those cell types when co-cultured and then deconstructed. I guess it's a good confirmation, but I found myself wanting to know if co-culture enabled these cell lines to become even more like their physiological *in vivo* counterparts than when the lines are induced and cultured separately in 2D format. In other words, do the authors have the negative control data (transcriptional analysis of each cell line, induced separately as monolayers) and/or positive control data (embryos)? Was there any difference in the transcriptional signatures in correctly organized and non correctly organized structures?

Author's reply:

We thank the reviewer for the suggestion to analyze 2D-mono-vs-3D-co-culture reprogramming in this context more closely, which significantly improved evaluations of experimental outcomes. We performed additional reprogramming and scRNA-Seq experiments, in which the conversion of ES-to-iTS cell fate and ES-to-iXEN cell fate was performed in 2D monoculture, according to published reprogramming protocols (Kaiser et al., 2020⁷; Niakan et al., 2013⁵). We found that both, iTSC and iXEN, clustered in close proximity to the transcriptional signatures of their 3D-reprogrammed counterparts. A direct comparison of enrichment levels did however reveal higher enrichment with their natural equivalents in 3D reprogramming for both, VE- and Epi-like cells, compared to lower enrichment

levels for reprogramming in traditional 2D mono-cultures (Fig 2G). In contrast to this, ExE-like cells obtained from 3D co-culture reprogramming displayed lower enrichment in natural ExE signatures, compared to their 2D mono-culture induced equivalents. Regarding comparison to positive control data (embryos), we merged and compared our scRNA-Seq with published datasets of a variety of developmental stages of murine embryogenesis, as proposed by Reviewer 1. The comparison of 2D mono- vs 3D co- culture reprogramming can now be found in Figure 2F, G, Supplementary Fig. 2J and Supplementary Table 5. Textual changes describing the 2Dvs3D reprogramming comparison and the merged dataset with murine embryos at different developmental stages can be found in line 185 – 223.

Related query: can ES, XEN, and TS cells be derived from these structures? Is there functional evidence that these cocultured cell lines adopt the expected cell fates (and not just transcriptional evidence)?

Author's reply:

We agree with the Reviewer's comment, that it is important to study the possibility of stem cell derivation from RtL-embryoids, as such cells generated by reprogramming in co-culture with naturally neighboring cell fates could provide a novel and possibly advanced approach to cell fate conversion.

We were able to isolate ESCs and iTSCs by outgrowth derivation in 2i/LIF ESC medium or TS Medium supplemented with FGF4/Heparin, respectively. These cell lines proliferated stably in culture up to passage 20 and beyond, retaining ES/TS morphology and showing continuous expression of ES/TS markers, as detected by IF staining (Figure S2K, L, M and N). We were however unable to stably derive iXEN cells from the aggregates, albeit testing different, published, derivation protocols (Niakan et al.,⁵ Lin et al., 2016²²). Respective textual changes can now be found in Line 223 – 230.

Could the authors provide evidence that CD40 is lineage-specific in this context?

Author's reply:

We agree with the reviewer's notion, that published lineage-specificity in murine embryos might not necessarily be reflected in RtL-embryoids. We followed the protocol introduced by Rugg-Gunn et al., 2012¹⁰, who demonstrated that CD40 can be used as a marker for the isolation of ExE cells from murine embryos, as long as cells of the Epiblast are excluded by detection of different markers. Similar to this published approach, we were able to isolate a CD40+ population of ExE-like cells, in combination with exclusion of GFP+ Epi-like cells and mCherry+ VE-like cells. We do however agree that this successful isolation of CD40+ ExE-like cells, does not necessarily indicate lineage-specificity of the marker and changed the textual passage within line 149 - 153 accordingly, which now reads: "Staining against CD40 in combination with GFP-signal-based exclusion of Epi-like cells allowed to separate CD40+ ExE-like cells from the remainder of the embryo-like structures, as described for early murine embryos¹⁰. Together with GFP and mCherry, this labeling allowed for the identification of VE-like cells (mCherry+/GFP-/CD40-), Epi-like cells (GFP+/mCherry-/CD40-), and ExE-like cells (CD40+/GFP-/mCherry-)."

Could the authors comment on how the proportion of synthetic embryo-like structures with asymmetric AVE-like zone compares with asymmetry of such structures created by other protocols (in other publications)?

Author's reply:

We agree with the Reviewer notion, that it would be important to place specific events described in this study in context with protocols for the induction of stem-cell-based embryo models. We performed IF stainings against additional DVE/AVE markers and were able to correlate expression of LEFTY1 within the EmVE with the DVE/AVE markers EOMES and OTX2 (Figure 3 F, G and H). In order to put this observation of an induction of a DVE/AVE-like zone, in context with asymmetry observed in other publications, we quantified LEFTY1+ in either distal-, transition- or anterior- positions, revealing the vast majority of LEFTY1+ cells to be located in distal- and transitioning- positions (Fig. 3 I and J). Expression of LEFTY1 in an anterior position was only observed in a minority of RtL-embryoids. As such, RtL-embryoids presented in this study seem to arrest or slow down in their developmental

potential and do not progress to a completely established anterior-posterior-axis. They therefore differ from other related protocols like e.g. Amadei et al., 2021³, in which the embryo-like structures complete AP-identity formation and progress to e.g. EMT. We therefore reduced our initial claim of “AVE symmetry breaking” to the induction of a DVE/AVE-like population within cells of the EmVE-like compartment (Line 257 – 272).

Related issue: the evidence of embryo patterning is quite superficial. Could the authors use standard assays for assessing embryo patterning (not single cell analysis in deconstructed structures, but in situ with patterning markers used by embryologists). Does T come on opposite the LEFTY1? Is there a gradient of phospho-MAPK in the EXE-like zone?

Author’s reply:

We agree with the reviewer’s comment that the observations and conclusions made from analysing the scRNA-Seq should be correlated with evaluation of patterning marker expression, as done by embryologists. Therefore, we performed additional assays for assessing embryo patterning in order to provide structural validation of the patterning observed in our scRNA-Seq dataset. Apart from a comprehensive analysis of changes in the pluripotency states of Epi-like cells, we were able to detect a gradient of phospho-MAPK in the ExE-like zone, similar to patterns observed in murine embryos between E5.5 – E6.5 (Corson et al., 2003⁶¹). RtL-embryoids displayed highest intensity of phospho-MAPK in the most distal region of the ExE-like compartment, reminiscent of FGFR independent pERK signaling in the ectoplacental cone emerging from the DiExE, and in ExE-like cells lining the Epi-like compartment, as for FGFR dependent pERK signaling in the PrExE. We thank the reviewer for the suggestion, as the proposed staining for pERK supported our scRNA-Seq based observation of a more differentiated character of ExE-like subcluster 2 and the stem cell characteristics of ExE-like subcluster 1. Additionally, comparison of transcriptional profiles of 2D-mono vs 3D-co-culture induced reprogramming proved to be very useful in analysing this bi partite character of the ExE-like compartment, as FGF4 and Heparin supplemented 2D reprogrammed iTSCs clustered closely to ExE-like subcluster 1, which also showed highest expression of the FGF4 receptor FGFR2. The respective data is now included in Figure 5 F, G, H and Supplementary Fig. 5C. Textual changes to the manuscript describing these observation can now found in line 390 – 414.

Regarding expressing of T on the opposite side of LEFTY1 expressing DVE/AVE-like cells, we did not detect expression of T in RtL-embryoids by IF staining and,

supporting this observation, only two cells in our scRNA-Seq dataset were T+. Considering that the vast majority of DVE/AVE-like zones was detected in a distal or transition position and scRNA-Seq of anterior-, transition- and posterior signatures only displayed enrichment in anterior-signatures, we concluded, that the developmental stage represented by RtL-embryoids does not progress to a T+ EMT stage. We therefore added the following paragraph to the section describing the Epi-like compartment, in line 315 - 317:

“Of note, primed pluripotency factors *Foxa2*, *Cer1* and *T* were only detected in single cells of the Epi-like cluster, providing further evidence for the presumed delay in progression from naïve- to primed-pluripotency”

Could the authors please move the graphs shown in Supp Fig 2A and B to the main figures. This contains an important accounting of the outcome efficiencies critical to supporting the authors' claims

Author's reply:

We did move the graphs accounting the outcome efficiencies to main Figure 1, as proposed by the reviewer. They can now be found in Figure 1 D and E.

Could the authors provide some kind of control for the uterine implantation analysis? As I recall, even beads of oil can cause decidualization, raising the possibility that these embryo structures aren't engaging a particularly unique outcome. What happens when periimplantation embryos are transferred to recipient females in the same manner?

Author's reply:

We do agree with the reviewer's comment that the implantations observed are most likely the result of decidualization due to mechanical stimulus, as observed when introducing e.g. beads of oil. Following the notion of Reviewer 1, we removed the section dealing with uterine implantation analysis, as we agree that the model presented here does not form the tissues required for and mediating uterine implantation.

Minorpoints:

Paragraph 1 of Intro: “conceptus as a whole is surrounded by the parietal yolk sac, Reichert’s membrane, and a layer of trophoblast giant cells.” In fact, the extraembryonic tissues are part of the conceptus; they do not surround the conceptus. Moreover, Reichert’s membrane is comprised of the parietal endoderm and the trophoblast giant cells.

Author’s reply:

We thank the reviewer for this correction and changed the respective text in line 49 – 51): “The conceptus’ outermost tissues consist of the parietal yolk sac and the Reichert’s membrane, a layer comprised of the parietal endoderm and trophoblast giant cells.”

Paragraph 1 of Results, “In recent years, we and others demonstrated that ESC can be reprogrammed to bona fide TSC and XEN cells (Wamaitha et al., 2015; Kaiser et al., 2020).” Should include additional relevant references (e.g., Niwa et al., 2005).

Author’s reply:

We agree, that Niwa et al., 2005 is an important relevant reference and added this additional reference to this publication. Due to limitations in number of references required according to formatting guidelines and additional references added in the course of this major revision, we were, unfortunately, unable to add more references.

Did the authors mix up distal and proximal EXE definitions? I know many papers put it this way, but it seems like the EXE’s axis should be inverted with respect to the epiblast’s axis, such that proximal epiblast and proximal EXE are neighbors.

Author’s reply:

Again, we thank the reviewer for the correction, as we followed seemingly commonly misused terminology used in other publications. We therefore changed the distal and proximal ExE definitions accordingly in the section describing the ExE-like compartment (Line 355 – 357, line 368 – 371 and line 390 - 414).

Figure 1 B-F could be reorganized. It was quite confusing that some panels are vertical and some are horizontal. One possibility is to eliminate some of the redundancy of the various color combinations.

Author's reply:

We agree with the reviewers comment, that panel and color combinations were in some cases redundant and thereby confusing. In the course of our extensive changes to the manuscript we also focused on eliminating color combination redundancy in order to highlight the relevant points described in the respective sections.

Figure 1 would be improved by including the quantification here – how many embryo-like structures resulted in correctly localized lineage-specific markers?

Author's reply:

We followed the reviewers advise to include the quantification of correctly compartmented structures in Figure 1.

Figure 3F – the color combos are hard to see – could the authors please separate the channels and show at least LEFTY1 alone?

Author's reply:

As previously discussed, we made significant changes to Figure 3, providing further IF staining in different color combos to further support the claim of a DVE/AVE-like population. We additionally now provide LEFTY1/DAPI and LEFTY1/GFP/DAPI panels separately, to increase visibility.

Figure 5D needs a scale bar of expression level (violin plot better)?

Author's reply:

We thank the reviewer for this correction and introduced the average expression by coloring the dots according to the average expression of the respective gene.

Figure 5E – which is proximal and which is distal?

Author’s reply:

The Featureplots shown in Figure 5E display the respective gene expression strength mapped on the UMAP representation of Figure 5A. Positions for ExE-like Subcluster 1 (PrExE-like) and ExE-like Subcluster 2 (DiExE) are therefore identical with the labels indicated in the larger UMAP in Figure 5A.

Figure 6A – I wasn’t clear what was meant by “ligand activity”

Author’s reply:

We agree with the reviewers comment, the term “ligand activity” is not really clear. We therefore added the following paragraph to the method section of the NichNet analysis, which can be found in line 760 - 765:

“NicheNet can address the question of which ligands produced by a sender cell are the most active in impacting gene expression in the receiver cell (i.e., ligand activity analysis)⁶². To achieve this, NicheNet assesses how well these ligands predict the examined changes in gene expression and ranks them according to this. After ligand ranking, NicheNet infers active ligand-target connections by looking for genes that are impacted in the receiving cell type and have a high possibility to be regulated by the prioritized ligands⁶²”

Why would implantations be expected to resemble E6.5 embryos after a week of in vivo development? What is observed at an earlier time point?

Author’s reply:

As previously discussed, we removed the section describing a possible implantation of the structures, as we agree with the reviewer’s notion, that such an implantation is most likely the result of decidualization, due to mechanical stimulus of the uterus. Following the advise of Reviewer #1 we removed the section from the manuscript.

Fig. 7D – was Cherry detectable?

Author’s reply:

See above

--

Reviewer #3 (Remarks to the Author):

The manuscript by Langkabel, Horne et al. describes a strategy for genetic modification of ESCs leading to induction of either TSC-fate or XEN-fate, and generation of embryo-like structures using mixtures of modified cells in a 3D co-culture. This approach would offer important advantages over existing, more complex methods for generating synthetic embryo-like structures. The authors largely rely on scRNA-seq to explore and validate the identities of cells of different compartments in the embryo-like structures. These analyses are based on a rather limited number of cells and biological replicates. However, in my opinion this is sufficient given the relative simplicity of the experimental model. The results are for the most part very clear and are in line with the conclusions made by the authors. The limitations of the study are also adequately described. Finally, the paper is very well written, and the quality of images is good. Altogether, I would recommend the paper for publication with the following minor remarks:

Figure 3D - differences in expression for some of the markers are not very clear. which of the differences are statistically significant?

Author's reply: We tested for difference in expression with the FindMarkers function (FindMarkers(Seurat_VE_sub, only.pos = TRUE, ident.1 = 1, min.pct = 0, logfc.threshold = log(0))) and could show that all genes of Figure 3D are significantly upregulated in the EmVE-like cluster, except for H2afy2.

	p_val	avg_log2FC	pct.1	pct.2
Nodal	4.45E-24	0.37445711	0.62	0.081
Fgf5	4.10E-19	0.3889263	0.43	0.038
Lefty1	6.84E-11	0.71885582	0.392	0.089
Sfrp5	4.33E-10	0.36141569	0.291	0.042
Sfrp1	3.21E-06	0.16158649	0.557	0.309
Lhx1	7.73E-05	0.19192183	0.228	0.072
Dkk1	0.0002615	0.42634096	0.759	0.653
Akr1c13	0.01040702	0.11332193	0.203	0.093
H2afy2	0.26663068	0.0446641	0.418	0.36

Figure 4C are these markers expressed in the other clusters besides Epi?

Author's reply:

We thank the reviewer for raising this issue. To respond to the reviewer comment we added Supplementary Figure 4A, in which the marker expression of pluripotency factors (Figure 4J) is shown in all main clusters of the RtL-embryoids. Additionally, we added a paragraph to the section describing pluripotency factor expression among all three compartments of RtL-Embryoids, in line 317 - 319.

Figure S3: The term “Top variable genes” is somewhat misleading, as it can also refer to the noisiness of expression. “30 most differentially expressed (DE)” would be more appropriate and this is also used in the main text.

Author's reply:

We agree with the reviewer, that the term “Top variable genes” is misleading and changed it to “differentially expressed genes” in Figures and Text.

Methods- Data quality control

The authors describe the following approach “We excluded the cells with less than 2500 expressed genes and less than 500,000 transcripts.” - What exactly is counted as a transcript? If this refers to aligned sequencing reads, the term should be changed accordingly.

Author's reply:

We agree with the reviewer, that the term “transcripts” is not precise and changed it to “aligned sequencing reads” in Figures and Text.

Reviewers' Comments:

Reviewer #1:

Remarks to the Author:

Dear authors,

The schematic describing the mouse conceptus and the embryo model (figure 1F) is important to understand what is being modelled. However, to be complete, please add the distal ExE, the Reichert's membrane, and the ecto-placental cone.

Line 51: As there seems to be confusion in the field, please add a sentence clearly mentioning that the interaction between the conceptus and the uterus is mediated by the blastocyst mural trophoctoderm that then gives rise to the primary TGCs, which then become covered by the Reichert's membrane and the parietal endoderm.

Line 55: Implantation occurs at the blastocyst stage through the interaction of a competent endometrium and an activated trophoctoderm. As such, embryo models that do not form these outer tissues, including the one of reference 2, cannot model the process of implantation. The fact that they induce a reaction of the uterus is irrelevant to the process of implantation. Many other manipulations can induce a reaction including a scratch (this was actually used in IVF clinics for a while), a drop of oil, or concanavalin-coated beads. Transferring a cerebral organoid into the uterus might induce some type of uterine reaction as well but I believe we would agree that this is irrelevant to implantation. The scientific process is fallible by nature and paved with errors, but these should be corrected. I call on your sense of responsibility to contribute to building solid foundations in the field of embryo modeling so that the scientific level can be further raised.

Line 107: What criteria did you use to assess the compartmentalization of the structures? Please mention it in the Material & Methods.

Lin 129: Please give an estimate of the % of structures that underwent fusion of the lumen. For a sake a clarity, you might want to add these percentages under the E5.25 and E5.5 in Fig. 1F.

Line 176: Please provide the GSEA enrichment scores directly in Fig. 2D.

Best wishes,

Nicolas Rivron

Reviewer #2:

Remarks to the Author:

The authors were highly responsive to the reviews and the manuscript greatly improved. The manuscript is now suitable for publication, although the authors may want to break some of the results into more paragraphs.

Reviewer #3:

Remarks to the Author:

The manuscript by Langkabel, Horne et al. describes a novel strategy for generating embryo-like structures using mixtures of genetically modified cells in a 3D co-culture. In the revised manuscript the authors have addressed all the minor issues I previously raised. Furthermore, the manuscript has been significantly improved by additional experiments confirming key findings and comparison of

scRNA-seq data with public data that provides additional insights and context. I have no further reservations and in my opinion the revised manuscript can be considered for publication.

REVIEWERS' COMMENTS

Reviewer #1 (Remarks to the Author):

Dear authors,

The schematic describing the mouse conceptus and the embryo model (figure 1F) is important to understand what is being modelled. However, to be complete, please add the distal ExE, the Reichert's membrane, and the ecto-placental cone.

Authors Reply:

We agree with the reviewers' comment and added the distal ExE, the Reichert's membrane, and the ecto-placental cone to the schematics in Figure 1f.

Line 51: As there seems to be confusion in the field, please add a sentence clearly mentioning that the interaction between the conceptus and the uterus is mediated by the blastocyst mural trophoblast that then gives rise to the primary TGCs, which then become covered by the Reichert's membrane and the parietal endoderm.

Authors Reply:

We agree that the process of implantation and the interaction between the conceptus and the uterus could be described in more detail. We therefore added the following sentence to the paragraph (Line 48 – 51):

"The implantation of the blastocyst into the uterus is mediated by the mural TE, that gives rise to primary trophoblast giant cells (TGCs), which subsequently become covered by the Reichert's membrane and the parietal endoderm, derived from the primitive endoderm."

Line 55: Implantation occurs at the blastocyst stage through the interaction of a competent endometrium and an activated trophoblast. As such, embryo models that do not form these outer tissues, including the one of reference 2, cannot model the process of implantation. The fact that they induce a reaction of the uterus is irrelevant to the process of implantation. Many other manipulations can induce a reaction including a scratch (this was actually used in IVF clinics for a while), a drop of oil, or concanavalin-coated beads. Transferring a cerebral organoid into the uterus might induce some type of uterine reaction as well but I believe we would agree that this is irrelevant to implantation. The scientific process is fallible by nature and paved with errors, but these should be corrected. I call on your sense of responsibility to contribute to building solid foundations in the field of embryo modeling so that the scientific level can be further raised.

Authors Reply:

As previously mentioned during the first revision process, we agree with the reviewers comments, that embryo-models such as the one of reference 2 do not form the outer tissues of a conceptus and can therefore not be regarded a suitable model for implantation. In response to the reviewers' comments during the first revision, we removed the respective data and experiments regarding "implantation" or rather uterine reaction from the manuscript to avoid any misconception in the field of embryo modeling. Additionally, we added the following paragraph to the respective section mentioned by the reviewer (Line 62 – 69).

“However, as implantation occurs at the blastocyst stage, mediated by the trophectoderm, which is not formed in the previously described embryo model⁹, the question has to be raised whether the observed implantations in fact represent uterine reactions, induced by mechanical stimuli of the uterus. Nevertheless, all of these embryo model systems can be used to recapitulate specific developmental events that occur during early embryogenesis and can potentially increase our knowledge in processes and diseases involved in embryogenesis. As a rapidly evolving field of research the potentials and limitations of such embryo-like organoids are currently actively discussed in the community.”

Line 107: What criteria did you use to assess the compartmentalization of the structures? Please mention it in the Material & Methods.

Authors Reply:

We added an additional paragraph to the Material & Methods section describing the criteria used to assess the compartmentalization of the structures (Line 616 – 625).

Line 129: Please give an estimate of the % of structures that underwent fusion of the lumen. For a sake a clarity, you might want to add these percentages under the E5.25 and E5.5 in Fig. 1F.

Authors Reply:

Considering the rare appearance of structures that underwent fusion of the lumen, we are unable to give a reliable estimate without reliable underlying data. We included the images as we did not want to exclude any data or observations from the manuscript. We explicitly stated that such seemingly more advanced developmental stage were rarely observed and should only be seen as an indication of the developmental potential of RtL-embryoids.

Line 176: Please provide the GSEA enrichment scores directly in Fig. 2D.

Authors Reply:

We realized that there was a wrong labeling in Fig. 2d, as we did not perform a GSEA analysis. In fact, we performed a GO term enrichment analysis, as already described in the methods and figure legends. The initial intention for the GO enrichment was to get an unsupervised view of the most enriched biological processes in the three main cluster of RtL-embryoids. All additional data for the GO enrichment analysis is specified in the Source Data file (GO enrichment analysis. Related to Fig. 2d) We thank the reviewer for the comment and corrected the labeling in Fig. 2d accordingly.

Best wishes,

Nicolas Rivron

Reviewer #2 (Remarks to the Author):

The authors were highly responsive to the reviews and the manuscript greatly improved. The manuscript is now suitable for publication, although the authors may want to break some of the results into more paragraphs.

Authors reply:

We thank the reviewer for the positive evaluation of the revised manuscript and the comments and suggestions during the first revision.

Reviewer #3 (Remarks to the Author):

The manuscript by Langkabel, Horne et al. describes a novel strategy for generating embryo-like structures using mixtures of genetically modified cells in a 3D co-culture. In the revised manuscript the authors have addressed all the minor issues I previously raised. Furthermore, the manuscript has been significantly improved by additional experiments confirming key findings and comparison of scRNA-seq data with public data that provides additional insights and context. I have no further reservations and in my opinion the revised manuscript can be considered for publication.

Authors reply:

We thank the reviewer for the positive evaluation of the revised manuscript and the comments and suggestions during the first revision.